# COSMOS: Compressed and Smooth Latent Space for Text Diffusion Modeling

**Viacheslav Meshchaninov\***
HSE University
Constructor University
vmeshchaninov@hse.ru

**Egor Chimbulatov**
HSE University
echimbulatov@hse.ru

**Alexander Shabalin**
HSE University
Constructor University
amshabalin@hse.ru

**Aleksandr Abramov**
SaluteDevices
andril772@gmail.com

**Dmitry Vetrov**
Constructor University
dvetrov@constructor.university

## Abstract

Autoregressive language models dominate modern text generation, yet their sequential nature introduces fundamental limitations: decoding is slow, and maintaining global coherence remains challenging. Diffusion models offer a promising alternative by enabling parallel generation and flexible control; however, their application to text generation is hindered by the high dimensionality of token-level representations. We introduce COSMOS, a novel approach to text generation that operates entirely in a compressed, smooth latent space tailored specifically for diffusion. This space is learned using an autoencoder trained simultaneously for token-level reconstruction and alignment with frozen activations from a pretrained language encoder, providing robust semantic grounding and enabling effective perturbation-based augmentations. Empirically, we demonstrate that text representations can be compressed up to $8\times$ while maintaining generation quality comparable to token-level diffusion models. Furthermore, increasing the latent sequence length allows COSMOS to surpass both diffusion-based and autoregressive baselines. We evaluate COSMOS on four diverse generative tasks including story generation, question generation, summarization, and detoxification and compare it with various generative paradigms. COSMOS achieves comparable or superior generation quality while offering more than $2\times$ faster inference. Code is released at GitHub.

## 1 Introduction

Autoregressive (AR) language models are a de facto gold standard for text generation [29, 30, 6]. By factorizing the probability of a sequence into a product of conditional token probabilities, they transform the global generation task into a series of local next-token prediction tasks that can be optimized efficiently via teacher forcing [42].

The same sequential factorization, however, creates several structural bottlenecks. First, decoding is inherently sequential: the time to produce a sequence grows linearly with its length because each token must await the completion of its predecessor. Second, the process suffers from exposure bias: a single early mistake contaminates the context for all subsequent predictions and cannot be corrected without restarting generation [32, 2, 11]. Third, maximizing local log-likelihood encourages the

---

Corresponding author: vmeshchaninov@hse.ru

39th Conference on Neural Information Processing Systems (NeurIPS 2025).

model to privilege fluency over factual accuracy, resulting in the well-documented phenomenon of hallucination [21, 13]. Finally, because decisions are made token-by-token, the model lacks an explicit global plan and therefore struggles to maintain long-range logical or narrative coherence [44].

In stark contrast, computer vision has been reshaped by diffusion models [12, 27], particularly by latent diffusion [33, 5, 4], which first compresses a high-resolution image or video into a compact latent representation and then performs the diffusion process in that space. Operating in a low-dimensional latent manifold reduces computational cost by orders of magnitude and enables breathtaking advances in image and video synthesis. Yet recent works [14, 37] also show that diffusion is highly sensitive to the geometry of the latent space: poorly designed latent representations can destabilize training and degrade sample quality, underscoring the need for principled mechanisms to construct *diffusable*[1] representations.

Motivated by these insights, we revisit the foundations of text representation. We hypothesize that the token-level encoding used in contemporary language models is heavily overparameterized for the purpose of sequence generation. Building on the successes of latent diffusion in vision, we pose the following question: *How far can we compress textual information into a compact latent space while still matching or even exceeding the generative fidelity of conventional token-level representations?*

To answer this question, we develop an autoencoder that maps text into a lower-dimensional space and then train a diffusion model directly in that space. We show empirically that naively minimizing token-reconstruction loss yields a brittle latent geometry that hampers diffusion, whereas introducing robustness- and smoothness-oriented objectives produces a well-behaved manifold conducive to high-quality diffusion synthesis. Our experiments demonstrate that with the right training regime, latent diffusion not only rivals but in several settings surpasses traditional token-level baselines.

These findings challenge the prevailing assumption that token-level autoregression is indispensable for language generation and position latent-space diffusion as a powerful alternative paradigm for future large-scale language models. Our key contributions are as follows:

- We propose COSMOS — a training recipe for a **CO**mpressed and **SMO**oth latent **S**pace, that allows to train a diffusion model in more compact latent space while matching the quality of token-level diffusion baselines.
- We demonstrate that modestly scaling the number of latent vectors enables latent-space diffusion to surpass both token-level diffusion and autoregressive models on an unconditional text generation task.
- On standard text-generation benchmarks, the proposed latent diffusion achieves up to $2\times$ faster sampling than conventional token-level diffusion models, while matching or slightly exceeding them in quality and diversity.

## 2 Related work

Early efforts to bring **latent** diffusion models into natural language generation treat the diffusion module mainly as a *pre-processor* that supplies conditioning vectors for an autoregressive decoder. LD4LG [19] trains a diffusion model on compressed hidden states from BART, then feeds the generated latent vectors into the BART decoder to produce text. PLANNER [47] adopts a similar two-stage recipe: a fine-tuned BERT encoder produces a 16-token variational latent code, a diffusion model refines that code, and a GPT-2 decoder generates text from this code. While both systems improve controllability, their heavy reliance on powerful autoregressive decoders makes it difficult to isolate and evaluate the intrinsic generative capacity of the latent-space diffusion itself.

A complementary line of work applies diffusion directly to the continuous embeddings of pretrained encoders. TEncDM [36] demonstrates that full-length BERT representations can be modeled with Gaussian diffusion and decoded into coherent text without an intermediate autoregressive step. Crucially, however, TEncDM retains the original sequence dimensionality, leaving open a question: *How aggressively can such representations be compressed before generation quality collapses?*

Our study closes this gap. We devise a training procedure that shrinks BERT-level representations by up to $8\times$ while preserving, and in some cases enhancing, their suitability for latent-space diffusion.

---

[1]Diffusable representations refer to latent spaces that permit effective modeling by a diffusion process.

Our study builds upon the approaches reviewed here. For additional context and a discussion of related topics please see Appendix B.

# 3 Preliminary

Diffusion models [12, 27] learn a data distribution by reversing a progressive noising process. Given a clean sample $z_0 \sim p_{\text{data}}$, the forward dynamic corrupts the input with Gaussian noise whose magnitude is controlled by a continuous time index $t \in [0, 1]$:

$$z_t = \sqrt{\alpha_t}\, z_0 + \sqrt{1 - \alpha_t}\, \varepsilon, \qquad \varepsilon \sim \mathcal{N}(0, I), \tag{1}$$

where the noise schedule $\alpha_t$ is monotonically decreasing with $\alpha_0 = 1$ and $\alpha_1 \approx 0$. A neural denoiser $z_\theta(z_t, t)$ with learnable parameters $\theta$ is trained to invert this corruption by minimising the denoising–score–matching objective

$$\mathcal{L}_{\text{DM}} = \mathbb{E}_{z_0 \sim p_{\text{data}},\, t \sim \mathcal{U}[0,1],\, \varepsilon \sim \mathcal{N}(0, I)}\left[\left\|z_0 - z_\theta(z_t, t)\right\|_2^2\right]. \tag{2}$$

At inference time the model is applied iteratively from $t = 1$ to $t = 0$, gradually transforming pure noise into a realistic sample.

## 3.1 Latent diffusion

Latent Diffusion Models (LDMs) [33, 4] improve both compute efficiency and quality by learning the diffusion process in a compact latent space rather than in pixel or token space. An autoencoder with encoder $E$ and decoder $D$ is first trained to achieve high-fidelity reconstruction, $\hat{\mathbf{w}} = D\big(E(\mathbf{w})\big) \approx \mathbf{w}$, where $\mathbf{z} = E(\mathbf{w})$ denotes the low-dimensional latent. Eqs. (1)–(2) yield a diffusion model trained in latent space whose network can be shallower, and faster than a counterpart operating in the original space, yet still achieves comparable perceptual quality after decoding $D(\mathbf{z})$.

In the remainder of the paper we adopt this framework for textual data: we compress contextualised text representations into the latent space, and train a diffusion model that operates within this space.

# 4 Methodology

## 4.1 Overview

**Frozen text encoder.** We initialise the pipeline with a pretrained contextual encoder $E_{\text{text}}$ (BERT-base [7] by default) that remains *frozen* during all subsequent training stages. For an input sequence of $L$ tokens $\mathbf{w} = (w_1, \dots, w_L)$, the encoder produces a matrix of hidden states $\mathbf{h} = E_{\text{text}}(\mathbf{w}) \in \mathbb{R}^{L \times d}$, where $d = 768$. Each row supplies a *semantically rich* representation of its corresponding token [38]. These representations provide a high-fidelity starting point for the compression stage.

**Compressor.** To distill the variable-length hidden states into a compact set of latents, we employ a Perceiver Resampler architecture [1]. It is a 12-layer transformer [39] in which every block replaces self-attention with cross-attention. Concretely, let $\mathbf{u} \in \mathbb{R}^{N \times d}$ be *a set of learnable vectors* initialized randomly and kept at a fixed length $N \ll L$. In each block, these vectors act as **queries** ($Q = \mathbf{u}W_Q$), while **keys** and **values** are formed by projecting a concatenation of $\mathbf{u}$ with the text encoder outputs ($K, V = [\mathbf{u}; \mathbf{h}]W_{K,V}$). Cross-attention therefore allows every vector $u_i$ gather information from the entire sequence of hidden states and from other vectors $\mathbf{u}$, gradually refining $\mathbf{u}$ into a semantically organized compressed representation. After the final block we obtain a *fixed-length* latent matrix $\mathbf{z} \in \mathbb{R}^{N \times d}$, which composes latent space for the diffusion process. For better interpretability of results we compress only along the sequence axis. Altering the embedding dimension $d$ would require architectural changes in the diffusion network itself, conflating encoder effects with diffusion capacity and obscuring the variables we aim to isolate.

**Latent normalization.** Before starting the Gaussian diffusion process, we estimate global mean and standard deviation $(\boldsymbol{\mu}, \boldsymbol{\sigma}) \in \mathbb{R}^{N \times d}$ on a held-out corpus, and normalize each latent feature so that it has zero mean and unit variance, $\mathbf{z} \leftarrow (\mathbf{z} - \boldsymbol{\mu})/\boldsymbol{\sigma}$. This step allows us to run variance-preserving diffusion process.

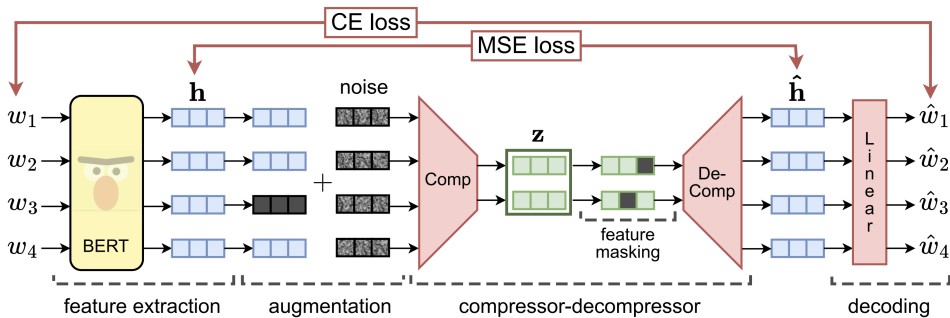

Figure 1: Overview of our training pipeline. A frozen BERT encoder extracts features, which are augmented before compression. A lightweight compressor–decompressor pair is trained with both token reconstruction (CE) and MSE objectives to produce compact and perturbation-resilient latent representations.

**Latent diffusion model.** We train a Gaussian diffusion model is space formed by $\mathbf{z}$, following Eqs. (1)–(2). Because $N$ is small, the diffusion model runs faster in this space.

**Decompressor.** Decompressor mirrors compressor in terms of architecture. It expands the fixed-length latent back to a sequence of $L$ vectors $\hat{\mathbf{h}} \in \mathbb{R}^{L \times d}$, capped at $L_{\max} = 512$ by default.

**Token predictor.** Finally, a linear projection followed by softmax converts each vector in $\hat{\mathbf{h}}$ into a probability distribution over the vocabulary, yielding the generated text.

In our work, we refer to the combination of the text encoder and compressor as the *encoder*, while the decompressor together with the token predictor constitutes the *decoder*.

### 4.2 Learning a compact text latent space

The previous Section 4.1 outlined our end-to-end pipeline: a frozen contextual encoder supplies token representations, a compressor distils them into a fixed-length latent matrix, and a diffusion model operates solely in this latent space. We now zoom in on the *compression* and *decompression* stages.

#### 4.2.1 How compact can text latents be?

Figure 2 reports token-level reconstruction accuracy on WIKIPEDIA dataset as a function of the latent sequence length $N$. The compressor outlined in §4.1 is trained using a token-level cross-entropy objective. Remarkably, a lightweight compressor equipped with only 12 transformer blocks is able to encode 512-token sequences into only $N = 16$ latents with 100% reconstruction accuracy, achieving a $32\times$ compression relative to initial 512 hidden states produced by BERT.

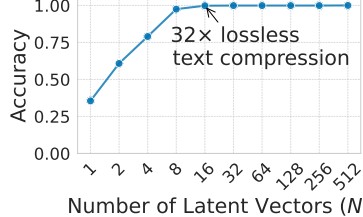

Figure 2: Token-level reconstruction accuracy on WIKIPEDIA (512 tokens) as a function of $N$.

These results paint an encouraging picture: if such a compact latent manifold can be modeled generatively, text generation could proceed in a space whose dimensionality is much smaller than that of the original token embeddings, promising dramatic speed-ups. However, as we show further, high reconstruction accuracy by itself does not imply that the latent manifold is suitable for generative modeling.

#### 4.2.2 Learning robust compact text representations

The reconstruction study (§4.2.1) reveals that aggressive text compression can be loss-free. However, for a robust diffusion generation compressed space should satisfy additional requirements. Empirically, we observe that if the latent manifold lacks smoothness and robustness (Section 5), a Gaussian diffusion model fails to sample latents that yield high-quality texts. To improve diffusability of

the latent space, we employ three complementary strategies to the autoencoder training. Figure 1 provides an overview of the proposed approach.

**MSE regularisation on encoder activations.** Alongside the cross-entropy loss between the original tokens and their reconstructions, we add a mean-squared error penalty between the frozen text encoder outputs $\mathbf{h}$ and their reconstructions $\hat{\mathbf{h}}$. This auxiliary objective forces the compressor to preserve the semantics carried by the contextual representations $\mathbf{h}$.

**Activation-space perturbations.** In order to teach compressor to extract additional features from $\mathbf{h}$ instead of just preserving its semantics, we apply perturb-and-recover training procedure. Concretely, we sample an augmented view $\mathbf{h}'$, pass it through the compressor–decompressor pipeline to obtain a reconstructed representation $\hat{\mathbf{h}}'$, and minimise $\mathrm{MSE}(\mathbf{h}, \hat{\mathbf{h}}')$, forcing both compressor and decompressor to remain invariant to the perturbation. Two perturbations are applied with equal probability within every batch:

(a) *Random masking*: $30\%$ of the vectors of $\mathbf{h}$ are set to zero, and
(b) *Gaussian noise*: after normalising $\mathbf{h}$ with the pre-computed statistics, we inject noise via $\mathbf{h}' = \delta\,\mathbf{h} + \sqrt{1 - \delta^2}\,\varepsilon$ with $\delta = 0.7$ and $\varepsilon \sim \mathcal{N}(0, I)$.

These augmentations force the autoencoder to tolerate partial information loss and promote smooth interpolation between adjacent latents (see Section 5.1).

**Latent-space augmentation.** We also apply augmentation directly to the latent matrix $\mathbf{z}$. Since the number of latent vectors $N$ can be an order of magnitude smaller than the token count $L$, masking whole latent vectors would annihilate too much information. Instead, during training we randomly zero out a fixed proportion $p$ of the *individual features* inside each latent vector. This fine-grained sparsification encourages neighbouring features within every latent vector to store redundant cues about one another, so that the representation remains intelligible even when a subset of features is excised, thereby making the manifold more robust to small latent perturbations..

## 5 Latent-space properties that facilitate diffusion training

This section embarks on an exploration of the text-latent manifold, asking which of its intrinsic features govern the performance of a diffusion model. All experiments reported here employ autoencoders that compress 128-token texts into 16 latent vectors. The autoencoders are trained on the WIKIPEDIA dataset, whereas the diffusion models are optimised on the smaller yet high-quality ROCSTORIES [23] dataset. Across all experiments, we generate 1 000 samples and report each metric as the average over these samples. Our experiments spotlight two indispensable qualities — *manifold smoothness* and *resilience to perturbations*. The remainder of the section introduces an analysis that quantifies both attributes.

### 5.1 Smoothness of the latent manifold

Once the autoencoder has been trained, each text $\mathbf{w}$ corresponds to a set of latent vectors $\mathbf{z}$ that can be decoded back into the original text, $\mathbf{w} = D(\mathbf{z})$. Nonetheless, the latent space is far from fully charted: broad regions contain vectors to which no text has ever been assigned, leaving the behaviour of the data density there unknown. Because our diffusion model captures the distribution of texts via the distribution of these latents, yet observes only a finite subset during training, its ability to generalize depends critically on how smoothly that density varies across the manifold. Put differently, the degree to which a locally estimated score function extends to unexplored territory is dictated by manifold smoothness. Inspired by PLANNER [47], we investigate this property by conducting the following experiment, which simultaneously evaluates performance at points the model does not encounter during training.

1. Select two random texts from the training corpus and encode them as latent vectors $\mathbf{z}^{(1)}, \mathbf{z}^{(2)} \in \mathbb{R}^{N \times d}$.
2. Form a linear interpolation of the endpoints $\mathbf{z}^{\mu} = \mu\,\mathbf{z}^{(1)} + (1 - \mu)\,\mathbf{z}^{(2)}$, where $\mu \in [0, 1]$ and mid–range $\mu$ values steer $\mathbf{z}^{\mu}$ into regions unseen during training.

3. Apply diffusion noise (Equation (1)) to get $\mathbf{z}_t^\mu$ at different time steps $t$ to measure how unseen regions influence diffusion at different noise levels.
4. Predict a clean latent $\hat{\mathbf{z}}_0$ from $\mathbf{z}_t^\mu$ and decode into text $\hat{\mathbf{w}}$.
5. Evaluate text plausibility with GPT–2 perplexity (PPL), averaged over $1\,000$ random endpoint pairs.

By analysing how perplexity (PPL) evolves with the interpolation coefficient $\mu$ and the diffusion timestep $t$, we obtain a fine-grained picture of manifold smoothness and model robustness far beyond the training support. Figure 3 shows that autoencoder trained only with cross-entropy (CE) loss and our robustness-oriented alternative behave identically when $\mu \to 0$ or $\mu \to 1$, confirming that both models successfully reconstruct train-time input latents. As we move away from the endpoints, perplexity rises for both encoders, but it escalates far more rapidly for the CE baseline — clear evidence that the latent manifold learned by our autoencoder is markedly smoother.

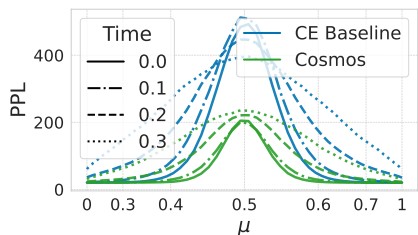

Figure 3: PPL of texts decoded from an interpolation of two latents for COSMOS and CE baseline.

## 5.2 Reducing the train–inference mismatch

We empirically observe that when a Gaussian diffusion model is trained in the text latent space, a persistent *train–inference mismatch* often emerges: the latent vector $\hat{\mathbf{z}}$ produced at sampling time can differ markedly from the latent vector the encoder would assign to its own decoded text, $E(D(\hat{\mathbf{z}})) \neq \hat{\mathbf{z}}$. This gap has two undesirable effects. First, the decoder becomes unreliable because it must interpret latent vectors contaminated by the diffusion model. Second, the diffusion model repeatedly feeds itself inputs it never saw during training, compounding errors over time. To enable high-quality text generation under this mismatch, both the decoder and the diffusion model must be robust to perturbations in the latent space. In the following sections, we demonstrate through experiments how our training strategy (§4.2.2) fosters such robustness, effectively decreasing train–inference mismatch.

**Decoder robustness.** To assess how well the decoder tolerates perturbations, we inject Gaussian noise into latent vectors of real texts: $\mathbf{z}_{\text{noised}} = \mathbf{z} + \sigma\,\boldsymbol{\varepsilon}$, where $\boldsymbol{\varepsilon} \sim \mathcal{N}(0, I)$, decode the perturbed vector to obtain $\hat{\mathbf{w}}$, and measure its BLEU score against the original text $\mathbf{w}$. Figure 4 shows that two components contribute most to decoder stability: (i) an explicit MSE loss on the decompressor's output (§4.2.2), which prevents the final-layer norm from exploding, a common effect observed when training solely with cross-entropy loss [8], and (ii) latent masking (§4.2.2), which forces the compressor to distribute information more evenly among latent features and remain resilient to partial dropout. Notably, the decoder supplemented with

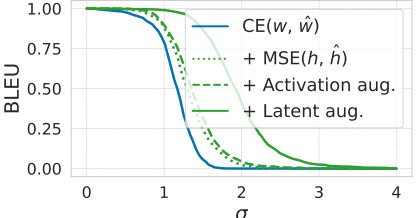

Figure 4: Decoder robustness to latent noising with sequential addition of training modifications.

described train-time modifications can reconstruct text almost perfectly even under substantial noise ($\sigma = 1$).

**Diffusion robustness during generation.** Next, we test the diffusion model's sensitivity to small perturbations introduced *mid-trajectory*. For a late generation step $t$, chosen from the second half of the reverse process, where latent representations are already semantically meaningful, we add Gaussian noise with magnitude $\nu$ and continue sampling. The deviation from the original trajectory is quantified by the MSE between the final estimates, $\|\hat{\mathbf{z}} - \hat{\mathbf{z}}^{\text{shifted}}\|_2^2$.

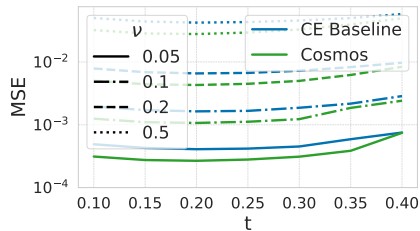

Figure 5: Evaluating diffusion model robustness under mid-trajectory noise injection.

As illustrated in Figure 5, the diffusion model trained in our latent space is visibly more stable.

**Direct mismatch measurement.** Table 1 tracks the train–inference mismatch, quantified as the MSE between $\hat{\mathbf{z}}$ and $E\big(D(\hat{\mathbf{z}})\big)$. Each proposed refinement tightens the gap. The consistent drop in error confirms that the enhanced training procedure makes the autoencoder more resilient to diffusion perturbations. Together, these results demonstrate that our training recipe simultaneously strengthens the decoder and the diffusion model, thereby shrinking the train–inference gap.

Table 1: Train–inference mismatch for different autoencoder objectives.

| Configuration | MSE $\big(\hat{\mathbf{z}}, E(D(\hat{\mathbf{z}}))\big)$ |
|---|---|
| $CE(w, \hat{w})$ | 0.721 |
| $+ \text{MSE}(h, \hat{h})$ | 0.619 |
| $+$ Activation aug. | 0.582 |
| $+$ Latent aug. | **0.499** |

# 6 Empirical study of latent representations for diffusion modeling

This section investigates how the design of autoencoder training strategies and the degree of latent space compression influence the quality of text generation via diffusion.

## 6.1 Empirical analysis of autoencoder training regimes

In this part of our study we assess the contribution of each proposed latent space augmentation to the final quality of the diffusion model. For the ablation study, we compress text from 128 BERT representations down to 16 latent vectors. All autoencoders are trained on the WIKIPEDIA dataset, while the diffusion models are trained on the ROCSTORIES [23] dataset.

**Evaluation metrics.** We evaluate the unconditional generation capacity of models with three complementary metrics that, together, capture textual quality, lexical diversity, and distributional fidelity. First, we report **perplexity (PPL)**, computed with GPT-2 LARGE [30]. Second, we quantify diversity using the score $\mathbf{div}(y) = \prod_{n=2}^{4} \frac{\text{\# unique } n\text{-grams in } y}{\text{\# } n\text{-grams in } y}$, where $y$ denotes the set of generated texts. Their, to ensure that the model does not reproduce the training dataset during the generation we evaluate the Memorization (**MEM**). Finally, because low perplexity can be achieved by simple or repetitive texts, we also compute the **MAUVE** score [28], which measures the distributional alignment between generated and reference texts and thus offers a broader view of generation quality. For every model we generate 1,000 samples and, when calculating MAUVE, draw an equally sized reference subset from the held-out test split. The entire evaluation procedure is repeated ten times with different random seeds, and we report the mean and standard deviation of the resulting scores across runs.

**Results.** Table 2 demonstrates that each proposed feature contributes significantly to the model performance. Remarkably, adding MSE penalty between the predicted and reference BERT activations alone boosts all quality metrics by about 50%. Subsequent augmentation

Table 2: Comparison of autoencoder training regimes on text generation quality. Features are added cumulatively from top to bottom. The gray-highlighted row indicates the selected configuration used in the final model.

| Configuration | MAUVE ↑ | PPL ↓ | Div ↑ |
|---|---|---|---|
| $\mathbf{CE(w, \hat{w})}$ | $0.294_{.036}$ | $71.5_{.9}$ | $0.298_{.001}$ |
| $\mathbf{+ MSE(h, \hat{h})}$ | $0.464_{.024}$ | $57.1_{.7}$ | $0.344_{.002}$ |
| **+ Random masking (rate)** | | | |
| 0.1 | $0.465_{.029}$ | $54.5_{.4}$ | $0.356_{.003}$ |
| 0.2 | $0.565_{.023}$ | $50.5_{.5}$ | $0.342_{.001}$ |
| 0.3 | $0.586_{.015}$ | $42.8_{.7}$ | $0.335_{.003}$ |
| 0.4 | $0.514_{.011}$ | $46.4_{.3}$ | $0.344_{.002}$ |
| 0.5 | $0.461_{.025}$ | $51.8_{.7}$ | $0.362_{.005}$ |
| **+ Gaussian noising ($\delta$)** | | | |
| 0.5 | $0.679_{.01}$ | $36.8_{.3}$ | $0.310_{.003}$ |
| 0.6 | $0.724_{.015}$ | $35.6_{.7}$ | $0.324_{.004}$ |
| 0.7 | $0.767_{.011}$ | $33.6_{.3}$ | $0.328_{.002}$ |
| 0.8 | $0.725_{.018}$ | $36.4_{.7}$ | $0.333_{.002}$ |
| 0.9 | $0.664_{.023}$ | $44.3_{.8}$ | $0.294_{.003}$ |
| **+ Latent dropout (rate)** | | | |
| 0.1 | $0.771_{.014}$ | $33.9_{.7}$ | $0.325_{.004}$ |
| 0.2 | $0.781_{.029}$ | $32.6_{.8}$ | $0.322_{.004}$ |
| 0.3 | $0.812_{.012}$ | $31.9_{.5}$ | $0.328_{.003}$ |
| 0.4 | $0.836_{.009}$ | $30.2_{.5}$ | $0.322_{.004}$ |
| 0.5 | $0.724_{.012}$ | $34.8_{.4}$ | $0.320_{.002}$ |

of the BERT activations yields a further leap, underscoring the value of intermediate contextual, semantically grounded representations. This stage rises MAUVE to 0.767, already achieving the performance obtained when the diffusion model is trained on the full-length, uncompressed BERT representations (0.767 vs. 0.762 for TEncDM [36]). Finally, dropping random features from la-

tent vectors pushes MAUVE beyond 0.83, lowers perplexity to 30.2, keeping diversity essentially unchanged.

These results provide a clear answer to the question posed in the introduction of this study: text representations can indeed be mapped into a more compact latent space, where a trained diffusion model performs comparably to traditional token-level counterparts without sacrificing the text generation quality. For all subsequent autoencoders we lock in the configuration highlighted in Table 2: the MSE penalty is kept, the random-masking rate is fixed at 0.3, Gaussian noising is applied with $\delta = 0.7$, and latent masking rate is set to 0.4.

Additionally, we explore the use of a variational prior in the latent space and observe that it offers no clear advantage. A thorough analysis is provided in Appendix D.1.

## 6.2 Impact of scaling the number of latent vectors

In this section, we examine how diffusion generation quality varies when varying the number of latent vectors $N$. We keep the training pipeline fixed according to the hyperparameters detailed in Section 6.1. We do not modify the embedding dimension $d$, because it would necessitate architectural modifications to the diffusion model, making it difficult to isolate the impact of latent space diffusability from changes in model capacity and structure.

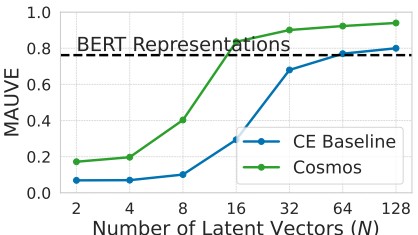

Figure 6: Impact of scaling $N$ on diffusion generation quality.

Figure 6 compares baseline cross-entropy compressor to the robustness-oriented compressor introduced in Section 4.2.2. We observe that for CE compressor decrease of $N$ leads to a rapid degradation in quality. In contrast, the proposed autoencoder maintains high generation quality even under substantial compression. It achieves an $8\times$ reduction in latent sequence length while surpassing the quality of uncompressed representations.

Table 3 provides further insights by comparing generation quality across different numbers of latent vectors $N$. Although configuration $N = 16$ matches the quality of BERT representations baseline, we observe a notable quality gap compared to the configuration with $N$=128: the MAUVE score decreases from 0.940 to 0.836, while perplexity increases from 26.3 to 30.2. However, adopting a less aggressive compression to $N$=32 (a $4\times$ compression) results in only minor quality degradation, still significantly outperforming the baseline. Based on these observations, together with results from Section 7, we recommend limiting latent compression to approximately $4\times$ to optimally balance quality and computational efficiency.

Table 3: Impact of scaling the number of latent vectors $N$ on unconditional generation quality.

| $N$ | MAUVE ↑ | PPL ↓ | Div ↑ |
|---|---|---|---|
| Source | 0.953 | 21.7 | 0.403 |
| BERT repr. | $0.762_{043}$ | $29.1_9$ | $0.295_{002}$ |
| 2 | $0.172_{018}$ | $109.5_9$ | $0.344_{004}$ |
| 4 | $0.197_{016}$ | $70.1_8$ | $0.354_{003}$ |
| 8 | $0.403_{007}$ | $51.9_7$ | $0.327_{002}$ |
| 16 | $0.836_{009}$ | $30.2_5$ | $0.322_{004}$ |
| 32 | $0.901_{008}$ | $27.3_4$ | $0.346_{004}$ |
| 64 | $0.923_{009}$ | $26.7_2$ | $0.347_{003}$ |
| 128 | $0.940_{011}$ | $26.3_4$ | $0.346_{004}$ |

## 7 Comparison across generative paradigms

We evaluate the performance of COSMOS on four distinct text generation tasks: unconditional story generation (ROCStories [23]), text summarization (XSum [25]), detoxification (ParaDetox [17]), and question generation (SQuAD2.0 [22]). For unconditional story generation we employ metrics discussed in Section 6.1. For text summarization and question generation, we report BERTScore (BS) [46]. The detoxification task is evaluated using BLEU-4 (BLEU). We further assess models' efficiency by comparing inference time for unconditional generation on sequences with 128 and 512 tokens. Additional metrics and implementation details are provided in the Appendix D.2.

**Baselines.** We benchmark our proposed method against a diverse set of generative paradigms. These include Gaussian diffusion on embeddings (DiffuSeq [10], SeqDiffuSeq [45], AR-Diffusion [43]),

Table 4: Comparison with autoregressive and diffusion baselines across four generative tasks. Inference time (in seconds) is reported for sequence lengths of 128 and 512 tokens. The best-performing scores are shown in **bold**, while the second-best scores are underlined.

| Method | ROCStories | | | | XSum | ParaDetox | SQuAD2.0 | Time (s) | |
| --- | --- | --- | --- | --- | --- | --- | --- | --- | --- |
| | MAUVE ↑ | PPL ↓ | Div ↑ | Mem ↓ | BS ↑ | BLEU ↑ | BS ↑ | $L = 128$ | $L = 512$ |
| Source text | 0.953 | 21.7 | 0.403 | 0.365 | — | — | — | — | — |
| GPT2 | 0.789 | 20.5 | 0.252 | 0.455 | 0.690 | 0.677 | 0.680 | **1.2**$_{.1}$ | 42.8$_{.1}$ |
| GPT Neo | 0.720 | **19.9** | 0.258 | 0.469 | 0.621 | 0.610 | 0.665 | 2.6$_{.1}$ | 45.2$_{.2}$ |
| AR-Diffusion | 0.066 | 41.8 | 0.101 | 0.540 | 0.568 | 0.647 | 0.569 | 226.4$_{.2}$ | 602$_{2.1}$ |
| DiffuSeq | 0.086 | 50.5 | 0.124 | 0.516 | 0.588 | 0.679 | 0.563 | 215.9$_{1.2}$ | 1565$_{.7}$ |
| SeqDiffuSeq | 0.103 | 29.3 | 0.137 | 0.663 | 0.617 | 0.688 | 0.574 | 100$_{.5}$ | 601$_{.5}$ |
| TESS | 0.061 | 22.4 | 0.163 | 0.550 | 0.627 | 0.693 | 0.667 | 1441$_{.2}$ | 5984$_{.0}$ |
| SEDD | 0.598 | 70.8 | 0.336 | 0.325 | 0.576 | 0.666 | 0.443 | 15.0$_{1.1}$ | 60.3$_{1.0}$ |
| LD4LG | 0.716 | 30.6 | 0.331 | 0.432 | 0.702 | **0.708** | 0.641 | 27.9$_{.5}$ | 102$_{.2}$ |
| TEncDM | 0.762 | 29.1 | 0.295 | 0.438 | 0.699 | 0.619 | 0.703 | 29.6$_{.1}$ | 180$_{.3}$ |
| COSMOS$_{N=16}$ | 0.836 | 30.2 | 0.322 | 0.394 | — | 0.654 | — | 5.8$_{.1}$ | — |
| COSMOS$_{N=128}$ | **0.940** | 26.3 | **0.346** | 0.383 | **0.704** | 0.694 | **0.708** | 35.1$_{.1}$ | **36.6**$_{.1}$ |

simplex-based diffusion (TESS [20]), masked diffusion (SEDD [18]), alternative latent diffusion baselines (LD4LG [19], TEncDM [36]), and autoregressive models (GPT-2 [30] and GPT-Neo [3]). All models operate within a comparable parameter scale, ranging from approximately 100M to 200M. To ensure fairness, all baselines were faithfully reimplemented and trained on generative tasks using training protocols closely aligned with those described by their original authors.

**Autoencoder setup across tasks.** To perform latent-space diffusion, we begin by training autoencoders on WIKIPEDIA dataset, varying the input sequence length. For tasks with relatively short input, such as unconditional story generation and text detoxification, we design two variants of the autoencoder that map 128-token input into either 16 or 128 latent vectors, resulting in the COSMOS$_{N=16}$ and COSMOS$_{N=128}$ configurations, respectively. In contrast, conditional generation tasks such as summarization and question generation require substantially longer input contexts. For these settings, we employ a third autoencoder configured to compress 512-token sequences into 128 latent vectors. Accordingly, we do not report results for long-context tasks for COSMOS$_{N=16}$, as sych aggressive compression fails to yield plausible outputs in these long-context scenarios. We adopt $N=128$ (4× compression) for long-context tasks as it strikes a favorable balance between generation quality and computational efficiency, as demonstrated in the scaling analysis in Section 6.2.

**Results.** As shown in Table 9, COSMOS$_{N=16}$ achieves strong performance relative to its closest latent diffusion baseline, TEncDM. It consistently outperforms TEncDM across most evaluation metrics, with the exception of unconditional perplexity, while offering significantly faster generation. Scaling up to a bigger latent representation, COSMOS$_{N=128}$ further advances generation quality. On ROCStories, it achieves a MAUVE score of 0.940, closely approaching the human reference score of 0.953, and substantially outperforming the best autoregressive baseline (GPT2), which reaches only 0.789. Although COSMOS$_{N=128}$ lags behind GPT2 in perplexity, this gap reflects an evaluation bias: perplexity inherently favors autoregressive models, particularly when assessed using the same decoding objective they were trained on.

In generation tasks that involve longer input contexts, COSMOS$_{N=128}$ consistently matches or slightly exceeds the performance of both diffusion-based and autoregressive baselines, while substantially reducing inference time. This efficiency stems from a core advantage of latent diffusion: unlike autoregressive decoding, where inference time grows linearly with sequence length, diffusion models operate with a fixed number of sampling steps. Although this cost is less favorable for short sequences, the benefits become increasingly pronounced as input length grows. This favorable scaling behavior makes latent diffusion as a promising direction for future research.

# 8 Experiments on OpenWebText

In this section, we present an empirical evaluation of our proposed method on the large-scale OpenWebText (OWT) [9] dataset.

## 8.1 Impact of Diffusion Model Depth

We train three diffusion models with varying depths of 12, 24, and 48 layers, which correspond to approximately 0.12B, 0.25B, and 0.5B parameters, respectively. For these experiments, the autoencoder and latent space configuration are held constant. We selected our best-performing autoencoder from preliminary experiments, which utilizes the largest latent space of 128 vectors for 128-token sequences.

Table 5: Scaling results on OpenWebText for 128-token generation. Increasing the number of layers in the diffusion model transformer improves generation quality and diversity. Best results in bold.

| Model Size | MAUVE $\uparrow$ | PPL $\downarrow$ | Div $\uparrow$ | Mem $\downarrow$ |
|---|---|---|---|---|
| 0.12 B | 0.849 | 97.6 | 0.492 | 0.135 |
| 0.25 B | 0.914 | 91.2 | 0.546 | **0.124** |
| 0.5 B | **0.923** | **89.7** | **0.554** | 0.125 |

The results presented in Table 5 clearly indicate a positive scaling trend. As the number of layers in the diffusion model increases, both generation quality and diversity improve. This demonstrates that increasing the capacity of the diffusion component is an effective strategy for enhancing performance, even when the autoencoder's architecture remains fixed.

## 8.2 Comparison on Long-Sequence Generation

We further assess our model's capabilities on the challenging task of generating 512-token sequences. We compare our model, COSMOS, in two configurations: without compression ($N = 512$ latent vectors) and with 2x compression ($N = 256$). These are benchmarked against TEncDM [36], a latent diffusion baseline, and GIDD [40], a strong discrete diffusion model. The results, presented in Table 6, offer several key insights into the advantages of our approach.

Table 6: Comparison with the GIDD masked diffusion model on OpenWebText for 512-token generation. Our model (COSMOS$_{N=512}$) shows significantly better quality and coherence.

| Model | MAUVE $\uparrow$ | PPL $\downarrow$ | Div $\uparrow$ | Mem $\downarrow$ |
|---|---|---|---|---|
| Source text | 0.968 | 23.2 | 0.465 | 0.036 |
| GIDD | 0.286 | 228.3 | **0.588** | 0.112 |
| TEncDM | 0.228 | 118.6 | 0.324 | **0.102** |
| COSMOS$_{N=256}$ | 0.263 | 84.3 | 0.368 | 0.156 |
| COSMOS$_{N=512}$ | **0.702** | **55.0** | 0.319 | 0.180 |

First, our primary model, COSMOS$_{N=512}$, demonstrates a significant performance gap over the discrete diffusion paradigm. It outperforms GIDD by a large margin, achieving a MAUVE score that is nearly three times higher (0.792 vs. 0.286) and a perplexity that is almost four times lower (55.0 vs. 228.3), indicating superior text quality and coherence.

Second, the results confirm the expected trade-off between compression and generation fidelity. While our compressed model, COSMOS$_{N=256}$, shows a degradation in performance compared to its uncompressed counterpart, it still outperforms TEncDM, a baseline that works with full-length, uncompressed textual representations. This result powerfully highlights the efficiency of our latent diffusion framework for text generation.

# 9 Conclusion

In this paper, we present COSMOS, a novel approach to text generation that shifts the focus from high-dimensional token-level representations to a compact latent space tailored for diffusion modeling. By learning a compressed and smooth latent representation through carefully designed autoencoder objectives, COSMOS enables high-quality generation while reducing inference time. Our empirical results show that, COSMOS matches or outperforms traditional token-level diffusion and autoregressive baselines across diverse tasks. Our findings challenge the dominance of token-level models and highlight latent diffusion as a promising direction for building fast, high-quality language models.

# Acknowledgements

The work was supported by the grant for research centers in the field of AI provided by the Ministry of Economic Development of the Russian Federation in accordance with the agreement 000000C313925P4E0002 and the agreement with HSE University № 139-15-2025-009. This research was supported in part through computational resources of HPC facilities at HSE University.

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

# Appendix

# A  Dataset descriptions

**ROCStories**  This dataset [24] consists of five-sentence stories and serves as a well-established, small-scale, unconditional benchmark in language diffusion research. It contains a total of 98,161 instances, of which 88,161 are used for training, 10,000 for validation.

**Wikipedia**  For large-scale experiments, we use the English Wikipedia subset from the ROOTS corpus [15]. The resulting dataset comprises 8.6 million sequences, with 50,000 held out for validation.

**XSum**  This dataset [26] is used for abstractive summarization and comprises 204,000 BBC news articles paired with summaries covering diverse topics (e.g., sports, politics). It includes 204,045 training instances, 11,332 validation instances, and 11,334 test instances. We truncate input articles to 512 tokens and limit reference summaries to 50 tokens.

**SQuAD2.0**  This question-answering dataset [31] requires identifying the text span that answers a given question or indicating that no answer is possible from the provided context. For generative modeling, we reverse the task: given a context and an answer, the model generates the corresponding question. The dataset contains 130,319 training and 11,873 test instances. Contexts are truncated to 512 tokens.

**ParaDetox**  For small-scale conditional generation experiments, we use the ParaDetox dataset [17], which comprises 19,766 pairs of toxic and neutral comments. Both context and target sequences are truncated to 40 tokens.

# B  Additional Related Work

In addition to the works discussed in the main paper, a significant body of research has explored diffusion models for text generation. These can be broadly categorized as follows.

## B.1  Gaussian Diffusion on Embeddings

One line of work focuses on applying Gaussian diffusion directly to word embeddings. Models like DiffuSeq [10] and SeqDiffuSeq [45] are designed for sequence-to-sequence text generation tasks. DiffuSeq employs a partial noising strategy and is trained end-to-end in a classifier-free manner. SeqDiffuSeq utilizes an encoder-decoder Transformer architecture and incorporates self-conditioning and an adaptive noise schedule to enhance performance. AR-Diffusion [43] integrates autoregressive principles into the diffusion process to better capture the sequential dependencies in language, allowing tokens on the right to depend on the generated tokens on the left. This is achieved by using a dynamic number of denoising steps that varies with the token's position.

## B.2  Simplex-Based Diffusion

Another direction explores alternative spaces for the diffusion process. TESS [20], for instance, applies a fully non-autoregressive diffusion process on the logit simplex space. This approach, combined with a novel self-conditioning technique, demonstrates strong performance on various natural language generation tasks while requiring fewer diffusion steps.

## B.3  Masked Diffusion

Masked-token diffusion models adopt a BERT-style prediction objective with iterative unmasking, enabling fast and parallel generation. For natural language, SEDD [18] established a connection between score-based models and ELBO maximization, demonstrating the effectiveness of the approach. Subsequent work, such as simple absorbing-mask diffusion with a clean training recipe, has successfully narrowed the perplexity gap to autoregressive language models and supports flexible semi-autoregressive decoding [34]. The quality of generation is also significantly boosted by various inference-time techniques. These include remasking for iterative correction [41], discrete guidance for improved controllability [35], and hybrid methods that expand self-correction capabilities [40].

### B.4 Latent Diffusion

In the realm of latent diffusion, several alternative baseline models have been proposed. LD4LG [19] learns a continuous diffusion model in the latent space of a pretrained encoder-decoder model. This method samples continuous latent representations which are then decoded into natural language, proving effective for both unconditional and conditional text generation. Similarly, PLANNER [47] adopts a two-stage recipe where a fine-tuned BERT encoder produces a variational latent code, which is then refined by a diffusion model before being translated into text by a GPT-2 decoder. TEncDM [36], on the other hand, operates in the space of pretrained language model encodings, which, unlike embeddings, integrate contextual information.

## C  Implementation details

### C.1  Latent diffusion pipeline

With a diffusable latent manifold in hand (§4.2.2), we now describe the diffusion model that operates entirely within this space. Our setup follows Eq. (2): we train a denoiser that receives a noisy latent $\mathbf{z}_t$ at diffusion time $t \in [0, 1]$ and predicts the corresponding clean latent $\mathbf{z}_0$. The loss is the simple mean-squared error $\|\mathbf{z}_0 - z_\theta(z_t, t)\|_2^2$, where $z_\theta$ shares parameters across all time-steps. In line with earlier work [19, 36], we employ *self-conditioning*: on 50% of training updates the model is fed its own previous estimate of $\mathbf{z}_0$ as an additional input. This iterative refinement enables the model, at inference time, to reuse its past predictions, producing markedly crisper and more coherent text. We adopt the noise schedule introduced by TEncDM [36]: $\alpha_t = \frac{1}{1+\tan(t\pi/2)^2 \cdot d^2}$, where parameter $d$ controls he rate at which noise is injected during the diffusion steps. At inference time we use Euler solver with 200 uniform steps. The denoiser itself inherits the TEncDM [36] architecture. It is a 12-layer Transformer with 12 attention heads and a hidden size of 768. In total, the network comprises roughly $\approx 130$M parameters.

**Conditional generation.** For sequence-to-sequence generation, we train a diffusion model in a conditional setting. The model learns to reconstruct a noisy latent representation of the target text, conditioned on the latent representation of the source text generated by our autoencoder. Consequently, the source text's vector representation is also leveraged in a compressed format, enhancing both training and inference efficiency. The diffusion model is conditioned on the source latent, which is first processed by a 12-layer transformer encoder. This encoder extracts relevant features and derives a suitable conditioning representation from the source latent. These encoded source representations are then injected into the diffusion model using cross-attention mechanisms. For conditional tasks, we initiate the training of our diffusion model by initializing its denoiser weights with those pretrained on an unconditional task.

### C.2  Training and inference configuration

All models are trained on 8 NVIDIA A100 GPUs. Detailed training configurations and approximate durations for both $\text{COSMOS}_{N=128}$ and $\text{COSMOS}_{N=16}$ are provided in Table 7.

### C.3  Inference time benchmarking configuration

We measure the generation time for a batch of 512 samples per model on a single NVIDIA A100 GPU using the `bfloat16` data type. Inference settings for all baseline models follow the default configurations of their respective repositories, except for SEDD, where we adopt 32 diffusion steps in accordance with the original paper's setup for matching autoregressive quality. We report the mean and standard deviation across five independent runs.

## D  Additional autoencoder training analysis

### D.1  Marginal utility of variational prior in text autoencoder

In this section, we explain why incorporating a variational prior does not significantly enhance the performance of our text autoencoder.

| | ROCStories | Wikipedia | XSum | SQuAD2.0 | Paradetox |
|---|---|---|---|---|---|
| Optimizer | | | AdamW | | |
| Learning Rate | | | 2e-4 | | |
| $(\beta_1, \beta_2)$ | | | (0.9, 0.98) | | |
| Warmup Steps | | | 1000 | | |
| Learning Rate Schedule | | | Constant | | |
| Weight Decay | | | 0.01 | | |
| Gradient Clipping | | | 1 | | |
| EMA Decay | | | 0.9999 | | |
| Batch Size | | | 1024 | | |
| Training Steps | 200k | 500k | 100k | 100k | 10k |
| Max Seq Length | 80 | 512 | 64 | 64 | 40 |
| Max Context Length | — | — | 512 | 512 | 40 |
| Sampling steps | | | 200 | | |
| Schedule parameter | 5 | 3 | 7 | 7 | 9 |
| Autoencoder Training Time | — | 14h | — | — | — |
| Diffusion Training Time | 1d 5h | 2d 12h | 1d 1h | 1d 9h | 14h |

Table 7: Training details for COSMOS across different datasets.

In a classical Variational Autoencoder (VAE), the encoder maps each input to the parameters of a Gaussian latent distribution, *i.e.*, a mean vector $\mu$ and a (diagonal) variance vector $\sigma^2$. Training minimises a reconstruction term together with the Kullback–Leibler (KL) divergence between the encoder distribution and an isotropic prior $\mathcal{N}(0, I)$. For a latent matrix $z \in \mathbb{R}^{N \times d}$, the KL term is

$$D_{\mathrm{KL}}\big(\mathcal{N}(z; \mu, \sigma^2) \,\|\, \mathcal{N}(0, I)\big) = \tfrac{1}{2} \sum_{i=1}^{N} \sum_{j=1}^{d} \big(\mu_{ij}^2 + \sigma_{ij}^2 - \log \sigma_{ij}^2 - 1\big).$$

Our text autoencoder is trained with the following objective:

$$\mathcal{L} = \mathrm{CE}(w, \hat{w}) \,+\, \mathrm{MSE}(h, \hat{h}) \,+\, \beta \, D_{\mathrm{KL}}\big(\mathcal{N}(z; \mu, \sigma^2) \,\|\, \mathcal{N}(0, I)\big),$$

where $\mathrm{CE}(w, \hat{w})$ is the token-level cross-entropy, $\mathrm{MSE}(h, \hat{h})$ matches reconstructed contextual representations, and $\beta \geq 0$ balances reconstruction fidelity against latent regularisation.

During training, the decoder receives a stochastic latent sample

$$z_s = \mu + \sigma \odot \varepsilon, \quad \varepsilon \sim \mathcal{N}(0, I),$$

so that the KL term nudges $(\mu, \sigma)$ towards the prior. Increasing $\beta$ strengthens this pressure, producing a smoother latent geometry at the cost of higher reconstruction error; decreasing $\beta$ does the opposite as encoder tends to push latents away from each other.

In our experiments, we observed that the performance is highly sensitive to the choice of $\beta$. Table 8 presents the results for different values of $\beta$.

The best perplexity is achieved at $\beta = 0.01$, but the gain over the baseline ($\beta = 0$) is marginal ($33.6 \to 32.6$). For larger $\beta$ the model collapses: at $\beta = 1$, perplexity explodes and MAUVE falls.

This limited improvement can be attributed to the decoder's robustness to Gaussian noise in the latent space. As illustrated in Figure 4, the decoder maintains reasonable performance even when sampling from the prior with $\sigma = 1$, indicating that the latent space is sufficiently robust without strong KL regularization.

Alternative regularisers, notably *latent masking*, yield larger and more robust improvements. We therefore omit the KL term in the final model and rely solely on latent masking to shape the representation space.

Table 8: Impact of the KL weight $\beta$ on diffusion generation quality. Arrows indicate the preferred direction.

| $\beta$ | MAUVE ↑ | PPL ↓ | DIV ↑ |
|---|---|---|---|
| 0 | **0.767** | 33.6 | 0.328 |
| 0.0001 | 0.733 | 39.6 | 0.329 |
| 0.001 | 0.764 | 33.3 | 0.328 |
| 0.01 | 0.765 | **32.6** | 0.326 |
| 0.1 | 0.658 | 39.6 | 0.329 |
| 1 | 0.011 | 677.1 | **0.587** |

## D.2 Additional details of comparison across generative paradigms

Table 9: Comparison with autoregressive and diffusion baselines across three generative tasks. The best-performing scores are shown in **bold**, while the second-best scores are underlined.

| Method | ParaDetox | | Xsum | | SQuAD2.0 | |
| --- | --- | --- | --- | --- | --- | --- |
| | BLEU ↑ | J-Score ↓ | R-1/2/L ↑ | BS ↑ | R-L ↑ | BS ↑ |
| GPT2 | 0.677 | **0.604** | 0.283/0.082/0.218 | 0.690 | 0.332 | 0.680 |
| GPT Neo | 0.610 | 0.492 | 0.231/0.045/0.171 | 0.621 | 0.245 | 0.665 |
| AR-Diffusion | 0.647 | 0.465 | 0.268/0.059/0.206 | 0.568 | 0.185 | 0.569 |
| DiffuSeq | 0.679 | 0.475 | 0.189/**0.130**/0.136 | 0.588 | 0.186 | 0.563 |
| SeqDiffuSeq | 0.688 | 0.486 | 0.286/0.067/0.213 | 0.617 | 0.194 | 0.574 |
| TESS | 0.694 | 0.587 | 0.317/0.116/**0.264** | 0.627 | **0.339** | 0.667 |
| SEDD | 0.666 | 0.001 | 0.200/0.033/0.138 | 0.576 | 0.086 | 0.443 |
| LD4LG | **0.708** | 0.580 | 0.303/0.100/0.246 | 0.702 | 0.211 | 0.641 |
| TEncDM | 0.619 | 0.496 | 0.319/0.107/0.253 | 0.699 | 0.323 | 0.703 |
| COSMOS$_{N=16}$ | 0.649 | 0.497 | — | – | — | — |
| COSMOS$_{N=128}$ | 0.694 | 0.554 | **0.328**/0.114/0.258 | **0.704** | **0.339** | **0.708** |

We also present extended results for our conditional generation tasks using a broader variety of evaluation metrics. For XSUM and SQUAD2.0, we report ROUGE scores [16], a standard metric that evaluates text quality based on n-gram overlap with reference outputs. For PARADETOX, we additionally include the J-score, which is defined as the product of *style accuracy*, *fluency*, and *content preservation*. Extended results are summarized in Table 9.

## E Limitations

There are several limitations to our study that point to promising directions for future work. First, jointly training the autoencoder and diffusion model remains an open research direction. Such an approach may significantly improve training efficiency. Second, the latent dimensionality $d$ is held fixed in all experiments, because changing $d$ would necessitate redesigning diffusion backbone; a systematic sweep of this hyper-parameter is therefore left to dedicated follow-up work. Finally, we report results with relatively small backbones of roughly 130M parameters to keep ablations rapid and compute budgets fair; scaling the architecture is an orthogonal engineering effort that is expected to reinforce the empirical trends observed here.

## F Societal Impact

Our research introduces an alternative modeling approach for language, centered on latent continuous diffusion. We hold the view that this method does not bring about significant new societal risks exceeding those already connected with current language models.

## G Generation examples

To give a qualitative sense of model behaviour, we first present representative unconditional generations: Table 10 shows random samples from COSMOS$_{N=128}$ and the TEncDM baseline on ROCStories, while Table 11 does the same for Wikipedia. We then move to the conditional setting, providing sequence-to-sequence outputs for three benchmarks: summaries for XSum in Table 12,

Table 10: Randomly generated samples for ROCStories dataset.

| **TEncDM** | **COSMOS**$_{N=128}$ |
|---|---|
| Amy wanted to buy a new dress for the dance! She shopped around for ten minutes. She tried on a dress that was too big. Amy loved that the dress was too grab for her size. She found that it was a little too long to wear! | Amy and her friend Sue were excited about sixth grade together. The three of them visited Amy's online shoe store. Amy's friend Beth arrived at the shoe store and Sue tried some dresses. The girls checked out and looked at all the collection. Amy felt like she was being an idiot. |
| Maggie was a nasty girl in school. When she moved to a small town, she had no Mae friends. After school, sheared out and made no friends. Sometimes they called her back, and they left her alone. She finally understood why she needed to be friends. | Jay was sitting at home bored. He was looking for an activity to keep him active. He decided to play a game of basketball. He played basketball for an hour. He was able to have a fun time that afternoon. |
| My car was becoming very worn out. I went to my mechanic to get checked out. He told me that I had a flat tire. I went to NCaven to get it fixed. I still made it to work in no time. | Sam was working late. He didn't have money to pay all his bills. He was having trouble getting paid. Sam decided to quit his work job. He was able to get another job to pay his bills. |
| My daughter is moving to a new school next year. She was nervous about moving to this new location. I am afraid of a leaker of all of her relationships. We figured it out and made some alterations on the team. This is where she will be graduating high school. | Debra hasn't written a check in weeks. She's been very stressed and frustrated with paying bills. Last week, she acted horribly during her check. The bank quickly ran a check - up. To her shock, she found out that she owed nearly a million in cash. |
| John went to the library. The librarian told him to read more books. John went to the booktore. He tried to read 6 books. He couldn't decide which to buy. | Tina never thought she needed her son to be cool enough to swim. So she begged him to do so while he was supposed to. So he practiced and swam all the time. But when it was time to go outside, he constantly got sore. Tina was too stressed to let him swim for 3 hours. |

detoxified rewrites for ParaDetox in Table 14, and question-generation examples for SQuAD 2.0 in Table 13.

Table 11: Randomly generated samples for the Wikipedia dataset using COSMOS$_{N=128}$.

The press said working conditions for the match did not have to be finalised. Even the organizers had planned for the date of the match to be 19 August 2017 to accommodate a crowd of some 500, 000 people watching the match.

Bob Hall served as Foundation founding president from 1990 to 1991. In 1993, Hansen was elected President of the College of Bymphoschipelgeons. Upon his retirement from this position, he also served as the chairman of the Scientific Advisory Committee of the College of Medical Surgeonsgeons ( IMS ). In 1996, publisher magazine selected Bob Hall as his candidate for the next U. S. presidential nomination for 2000 and 2001. He died October 26, 2013, at his home in Tampa, Florida from the causes of kidney failure at the age of 71, caused by a cerebral kidney failure in 1980. He is buried at the

He was excluded from the committee because Johnson refused to approve the text of the bill, and two senators refused to attend discussing the legal provisionss of the Seventh Amendment, but invited senior officials to hear it. Both the Congress House and the Weed Hermans challenged the committee's request to have the Seventh Amendment be argued in the Supreme Court's decision during the subsequent legislative hearings. Both attributed the premise premisel of Johnson's Amendment to internal law, which allocated the Congress to adopt executive decrees beginning in January 1699. The law, for its part, was however applied by the majority of the

Folk music typically contains a mix of German, Russian, Russian, Iranian, Egyptian, Norwegian, Turkish, Jewish and Tabad music, though typically the songs are from a specific religion or background. Icelandmar00edk, for instance, distinguish genderly distinguishes folk music from the multi - diverse genres, primarily rock and jazz.. Folk music is consequently not subegoaticly distinct from rock, which in turn, which consists primarily of music of folk including jazz, blues, nor rock, and industrial rock. Due to the diversity of genres in these genres, certain new genres of music also exist. Pandit music, such as

In 1974, Parker became the Director of the Office of Virginia's Chamber of Trade and Commerce, a position he would hold under increased pressure from his fellow Democratic opponent. He oversaw the J. Howard administration, which lasted for 11 years headed by William J. P honorter.

Table 12: Samples from XSum summarization dataset. Parts of the articles are omitted for brevity.

**Article:** Two-year-old Lane Thomas Graves had been playing in the sand near the resort's Seven Seas Lagoon when he was dragged underwater by the creature... The lighthouse has been installed near to where the attack occurred... A Disney spokesperson said they hoped the monument would spread awareness for the Lane Thomas Foundation, which also uses the lighthouse as its logo. Who is liable for alligator boy's death? "The lighthouse sculpture has been installed to help spread awareness of the Lane Thomas Foundation, which was established to provide assistance and support to families whose children need organ transplants," Walt Disney World said in a statement.

**Reference:** Walt Disney World has unveiled a lighthouse memorial for a young boy who was killed by an alligator while on holiday at the Florida theme park.

| **TEncDM** | **COSMOS**$_{N=128}$ |
|---|---|
| A statue of a Florida boy who died after being rescued from a beachqua-rina during water has been honoured by US television provider Disney. | Walt Disney Disney has unveiled a lighthouse statue in memory of a young boy who died when he was stabbed by an alligator at a Florida resort in Florida. |

**Article:** The Sky Blues currently play in Coventry's Ricoh Arena but had a long dispute with the stadium's previous owners... In a statement, Rugby Borough Council said its leader and the council's executive director and head of planning had met with Coventry City in March. "The club requested the meeting to understand how the council would deal with any planning application for potential stadium sites in the borough of Rugby," it said. It said the plans would need to be finalised by September to be included in the council's local plan, but added that a site had yet to be identified. Peter Ward, from Sky Blues Supporters' Consultative Group, said he was pleased to hear that things were "moving" with the club's search for a new home. "It's good that finally there is some evidence things

**Reference:** Planners in Rugby have revealed they have been in talks with Coventry City Football Club about building a stadium in the borough.

| **TEncDM** | **COSMOS**$_{N=128}$ |
|---|---|
| Coventry City Council says it is looking on whether potential plans for a 0̆0a33m Super League stadium in Coventry. | Coventry City fans say they will meet with the city council over a proposed move from the club club to a new stadium. |

Table 13: Samples from SQuAD2.0 question generation dataset.

**Context:** Temporal measurement has occupied scientists and technologists, and was a prime motivation in navigation and astronomy. Periodic events and periodic motion have long served as standards for units of time. Examples include the apparent motion of the sun across the sky, the phases of the moon, the swing of a pendulum, and the beat of a heart. Currently, the international unit of time, the second, is defined by measuring the electronic transition frequency of caesium atoms (see below). Time is also of significant social importance, having economic value (ẗime is money)̈ as well as personal value, due to an awareness of the limited time in each day and in human life spans.

**Answer:** Temporal measurement.

**Question:** What has been a prime motivation in astronomy and navigation?

| **TEncDM** | **COSMOS**$_{N=128}$ |
|---|---|
| What has a significant role in sur-veying units of time? | What have technologists filled time in astronomy? |

**Context:** In Book 11 of his Confessions, St. Augustine of Hippo ruminates on the nature of time, asking, Ẅhat then is time? If no one asks me, I know: if I wish to explain it to one that asketh, I know not.Ḧe begins to define time by what it is not rather than what it is, an approach similar to that taken in other negative definitions. However, Augustine ends up calling time a Ž01cdistentionŽ01d of the mind (Confessions 11.26) by which we simultaneously grasp the past in memory, the present by attention, and the future by expectation.

**Answer:** St. Augustine of Hippo.

**Question:** Who commented on the nature of time in Book 11 of his confessions?.

| **TEncDM** | **COSMOS**$_{N=128}$ |
|---|---|
| Who wrote " " nature of timefes-sions ? | Which philosopher ruminates about explaining the nature of time? |

Table 14: Random samples for the detoxification task on the ParaDetox dataset.

| | |
|---|---|
| **Input:** fucking imagine obama just put the hands up and keep a good distance between he and cruz . | |
| **Reference:** Imagine Obama just put the hands up and distanced himself with Cruz. | |
| **TEncDM** | **COSMOS**$_{N=128}$ |
| Ž018 obama just put the hands up and keep a good distance between him and cruz. | imagine obama just put the hands up and keep a good distance between he and cruz. |
| **Input:** you just summed up how fucking stupid politics is in one comment | |
| **Reference:** You just summed up how politics is in one comment. | |
| **TEncDM** | **COSMOS**$_{N=128}$ |
| you just sum up up how bad politics is in one comment | you just sum up up how how stupid politics is in one comment |
| **Input:** only churches give a crap it seems. | |
| **Reference:** only churches care it seems. | |
| **TEncDM** | **COSMOS**$_{N=128}$ |
| Only churches care bad it it seems.. | only churches give a thing it seems. |
| **Input:** second , maybe if you douches would read the damn edit summaries , this wouldnt happen . | |
| **Reference:** second , maybe if you would read the edit summaries , this wouldnt happen . | |
| **TEncDM** | **COSMOS**$_{N=128}$ |
| Ž01d, maybe if you would read the editmaries. | If you read the edit summaries, this wouldn't happen. |
| **Input:** what we witnessed here today was a fucking miracle, and i want you to acknowledge that. | |
| **Reference:** What we witnessed here today was a miracle, and I want you to acknowledge that. | |
| **TEncDM** | **COSMOS**$_{N=128}$ |
| What we witnessed here today was a miracle, and I want you to acknowledge that. | what we witnessed here today was a a miracle, and i want to acknowledge that. |

