# OpenReview forum: "Cosmos: Compressed and Smooth Latent Space for Text Diffusion Modeling"
_NeurIPS.cc/2025/Conference — NeurIPS 2025 poster_

### Official Review · Reviewer_QQwS · 2025-06-27

**Clarity:** 3
**Significance:** 2
**Originality:** 2
**Rating:** 4
**Confidence:** 3

**Summary:**

The paper studies the use of a learned continuous and compressed latent spaces for text generation. A large part of the paper is focused on how to regularize the latent space to improve its “diffusability” and on evaluations to understand how each of these regularizations affect the learned latent space and the subsequent performance by the latent diffusion model. One of the main benefits of the proposed method comes from running a generative model in a lower dimensional space (typically 4x and 8x smaller), which leads to faster sampling.

**Questions:**

In the diffusion model evaluations, what is the decoded sequence length (that is, L)? How exactly can the method be used to generate longer outputs than the ones obtained by decoding the fixed length latent variables to a fixed output size. (Related to the next question.)

I understand in this work the method is always trained on inputs of certain size L=512, and the latent space has L / c, where c is the compression factor >1. You could train this on sequences of varying lengths I suppose? By using not a fixed size latent space, but a latent space of size (L / c) x d, where L changes every batch.

I was not totally clear on how you are introducing context (prompt) in the method, for instance, for short or long context tasks.

Why are other baselines, for instance MDLM, not considered? Is it due to the scale of the networks used? (This work focuses on architectures in the 100M-200M range).

**Ethical Concerns:**

["NO or VERY MINOR ethics concerns only"]

**Final Justification:**

Please see discussion.

**Limitations:**

The paper includes a limitation section. See weaknesses above for some questions regarding other potential limitations of the method.

**Quality:**

3

**Strengths And Weaknesses:**

Strengths:
- The analysis and evaluations of the regularization techniques used are quite thorough and indicate that the regularizations used are quite relevant for the task at hand. I consider this to be quite nice, as learning the right latent space for these tasks (i.e. well suited for a diffusion model to operate in the latent space) is critical.
- The model gets faster inference without sacrificing performance.
- The fact that the method builds on continuous space diffusion models enables the use of tools developed in that space, such as distillation and other fast sampling techniques, which are not yet as developed for discrete diffusion models.

Weaknesses:
- Typical AR language models rely on low temperature sampling, which appears to be somewhat important for good performance. Moving to the continuous latent diffusion framework, it is unclear how this could be done. While there are techniques for low temperature sampling for diffusion models in continuous spaces, these are quite heuristic, and the final effect may be quite different to the one obtained when doing low temperature sampling for LLMs (the fact that the model operates on a latent space may exacerbate this).
- I don’t find the empirical evaluation of the final diffusion models entirely convincing regarding the performance of the method at scale. Models are on the smaller side (100M-200M parameters), trained on small-to-medium size datasets (ROCStories, XSum, etc.), and on fairly small latent spaces (16 x dim). For longer sequences, the 16 should be increased? Or would this approach be used in an autoregressive manner generating blocks?
- While I don’t consider this necessary, it could be very nice to have. One strength I mentioned above involves the use of distillation techniques that are only applicable in continuous space diffusion models. It would be great to see how these techniques interplay with the proposed method. This would be orthogonal to the paper’s contributions, but is aligned with the whole direction of increased efficiency. Maybe such techniques could give another boost in speed, which would be very nice.

---

> ### Author Rebuttal · Authors · 2025-07-30
>
> We sincerely thank the reviewer for the positive and encouraging feedback. We are pleased that you found our analysis of the regularization techniques thorough and relevant, and we appreciate your recognition of the efficiency gains and the advantages of building on continuous‑space diffusion models. Below, we address each point raised.
>
> > **W1.** On the analogue of temperature sampling in the continuous latent diffusion framework.
>
> Temperature sampling in autoregressive models allows for increasing or decreasing the confidence of token sampling, which in turn typically improves quality at the expense of diversity, and vice versa. In continuous Gaussian diffusions, there is indeed no direct concept of temperature sampling. However, we can propose a technique with an analogous effect. The work of TEncDM [1] shows that self-conditioning increases the network's confidence in its predictions. Based on this fact, we can control the prediction confidence using the following procedure to obtain the prediction $x_0$:
>
> $x_0^{t} = x_{\theta}(x_t, t, \emptyset) + w \cdot \bigg(x_{\theta}(x_t, t, x_0^{t+1}) - x_{\theta}(x_t, t, \emptyset)\bigg).$
>
> Here, the final prediction $x_0^t$ at step $t$ is an extrapolation from the unconditional prediction $x_{\theta}(x_t, t, \emptyset)$ in the direction of the self-conditioned prediction $x_{\theta}(x_t, t, x_0^{t+1})$, controlled by a weight $w$.
>
> A larger $w$ increases model confidence, leading to higher-quality generations, while a smaller $w$ promotes diversity. Our key finding here is that this trade-off can be finely controlled; by tuning $w$, it is possible to achieve substantial improvements in quality with only a minor decrease in diversity.
>
> | $w$   | MAUVE $\uparrow$ | PPL $\downarrow$ | Div $\uparrow$  |
> |-----|---------|--------|--------|
> | 0.9 | 0.781   | 33.5   | **0.326**  |
> | 1.0 | 0.836   | 30.2   | 0.322  |
> | 1.1 | 0.842   | 28.3   | 0.315  |
> | 1.2 | 0.845   | 27.4   | 0.315  |
> | 1.3 | **0.867**   | 27.3   | 0.314  |
> | 1.4 | 0.841   | 26.9   | 0.309  |
> | 1.5 | 0.843   | **26.3**   | 0.306  |
> | 1.6 | 0.842   | 27.3   | 0.306  |
>
> This represents a very useful practical feature of our approach. We appreciate you raising this point and will add these results to the final version of the paper.
>
> > **W2.** On empirical evaluation.
>
> Latent diffusion for text generation is a relatively new and under-explored area, as also noted by Reviewer Kj8c. Prior to developing a large‑scale model, it is crucial to validate the underlying hypotheses for training the latent space and the diffusion model at a smaller scale. This work accomplishes precisely that, laying the groundwork we consider essential for successful large‑scale development.
>
> To provide initial evidence of this scalability, we have taken the first steps on the OpenWebText dataset by investigating if quality improves when increasing the diffusion model's size, keeping other components fixed. By increasing the number of layers, we found that generation quality and diversity both improve significantly:
>
> | Model Size | $n_{layers}$ | MAUVE $\uparrow$ | PPL $\downarrow$ | Div $\uparrow$  | Mem  $\downarrow$ |
> |------------|----------|-------|------|-------|-------|
> | 0.12 B     | 12       | 0.849 | 97.6 | 0.492 | 0.135 |
> | 0.25 B     | 24       | 0.914 | 91.2 | 0.546 | **0.124** |
> | 0.5 B      | 48       | **0.923** | **89.7** | **0.554** | 0.125 |
>
> Additionally, we trained an autoencoder using our strategy without compression and a diffusion model with a sequence length of 512 on OpenWebText, for comparison with the discrete diffusion method GIDD, which was trained under identical conditions.
>
> | Model        | MAUVE $\uparrow$ | PPL $\downarrow$  | Div $\uparrow$  | Mem $\downarrow$  |
> |--------------|---------|--------|--------|--------|
> | Source text  | 0.968   | 23.2   | 0.465  | 0.036  |
> | GIDD         | 0.286   | 228.3  | 0.588  | 0.112  |
> | Cosmos       | 0.524   | 59.0   | 0.315  | 0.171  |
>
> These results are a strong positive indicator for scaling.
>
> Regarding generation length, we used a fixed number of latents for all texts in this work. Adapting the model for longer texts requires separate research as, you correctly noted, there are several viable strategies to explore
>
> > **W3.** On distillation of continuous diffusion model.
>
> We agree that exploring the distillation of latent text diffusion models is a fascinating direction for future work. However, we are cautious about the direct application of techniques from the image domain, as we are not aware of any work that has successfully distilled a continuous Gaussian diffusion model into a one-step generator for text. As you rightly point out, such an acceleration technique is orthogonal to our core contribution of learning a compressed latent space for diffusion. We believe our model will become a good framework for beginning research in this direction.
>
> > **Q1.** On sequence length.
>
> As we wrote in Section 7 in the paragraph "Autoencoder setup across tasks," the decoder sequence length (L) is 128 or 512, depending on the experiment. To generate sequences of arbitrary length, we follow the same procedure as in other works on text diffusion [1, 2], where the decoder reconstructs a sequence up to a maximum length with padding, and the output is then cut off after the first EOS token. Consequently, the decoder in our framework does not produce sequences that exceed the predefined maximum length.
>
> > **Q2.** On using not a fixed size latent space.
>
> We thank you for this insightful suggestion. To keep the core methodology focused, we employ a fixed latent space size in this work. We agree that using a variable number of latents for texts of varying lengths is a compelling idea for future exploration, though it requires additional research beyond the scope of this paper.
>
> > **Q3.** On introducing context in the method.
>
> To encode the context before feeding it into the diffusion model, we first pass it through our autoencoder, which generates a compressed representation. It is this representation that is subsequently supplied to the diffusion model via cross-attention. Depending on the task and context length, we use a model configured for either 512 or 128 tokens for this process.
>
> > **Q4.** On baselines.
>
> Our comparative analysis was designed to benchmark our method against key paradigms in text diffusion: Gaussian diffusion on text embeddings, simplex-based diffusion, masked diffusion, and latent diffusion. From the many masked diffusion models available, we selected SEDD as a prominent and established baseline. For a more direct and controlled comparison, we also benchmark against GIDD (as detailed in our response to W2), as it was trained under the same conditions and text length as our model. We considered MDLM but opted against it because its training on single-line concatenated texts produces fragmented, non-comparable samples. To avoid re-training a method specifically for our setup, we decided to use GIDD, which was trained in the exact same setup as our model.
>
>  $\newline$
>
> References:
>
> [1] Shabalin, Alexander, et al. "Tencdm: Understanding the properties of the diffusion model in the space of language model encodings" (2025).
>
> [2] von R{\"u}tte, Dimitri, et al. "Generalized interpolating discrete diffusion" (2025).
>
> ---
>
> We sincerely appreciate your constructive feedback, which will greatly assist us in improving the final manuscript. Please let us know if there are any remaining points that require clarification.

---

> > ### Comment · Reviewer_QQwS · 2025-08-06
> > **Thanks for the response**
> >
> > I thank the authors for their response, and for including the additional results with classifier free guidance (and others). I'll keep my acceptance score.

---

> > > ### Author Response · Authors · 2025-08-07
> > > **Thank you for your constructive engagement.**
> > >
> > > Thank you once again for your thoughtful review and for considering our rebuttal. We truly appreciate the time and expertise you dedicated to our submission.
> > >
> > > Your feedback prompted us to look beyond the initial scope of our work. We were particularly excited by the results on scalability and the new self-conditioning technique. We believe these additions not only validate our core hypothesis but also lay a solid foundation for future research in latent text diffusion, including promising avenues like distillation that you pointed out. We feel the paper now presents a much more complete and robust contribution.
> > >
> > > Thank you again for your constructive engagement.

---

### Official Review · Reviewer_ojHu · 2025-07-02

**Clarity:** 3
**Significance:** 2
**Originality:** 2
**Rating:** 4
**Confidence:** 3

**Summary:**

This paper introduces COSMOS, an approach for text generation that leverages a compressed and smoothed latent space specifically tailored for diffusion models. Using a combination of a frozen language encoder, a lightweight compressor-decompressor architecture, and robustness-promoting training objectives, the method aims to achieve efficient, high-quality text generation with latent spaces significantly smaller than token-level representations. COSMOS is evaluated on a range of text generation tasks (such as story generation, summarization, question generation, and detoxification) and compared against both autoregressive and diffusion baselines. The experiments demonstrate that COSMOS achieves comparable or superior text quality while providing substantial speed-ups in inference.

**Questions:**

Q1. What is the fundamental difference between your method and LD4LG?
Q2. Public datasets and methodologies must include citations or sources.
Q3. In Section 4.2.2, there are two consecutive ending punctuation marks in the last line. Please fix this issue.
Q4. In Section 6.2, you state: "It achieves an 8× reduction in latent sequence length while surpassing the quality of uncompressed representations." However, according to Figure 6, this phenomenon only holds for the comparison between N=128 and N=16, not universally. Recommend revising the wording to reflect this nuance.
Q5. Reproducible code and implementation details.
Q6. The evaluation metrics lack human evaluation results under few-shot settings.
Q7. Related work deserves a more detailed discussion.
Q8. Long-context experiments.
Q9. Direct experimental validation of exposure bias mitigation.

**Ethical Concerns:**

["NO or VERY MINOR ethics concerns only"]

**Final Justification:**

This paper introduces COSMOS, a diffusion model-based text generation method based on compressed and smoothed latent spaces. The core idea is to combine a frozen language encoder, lightweight compression–decompression modules, and robustness-oriented training objectives to significantly shorten the latent sequence length while maintaining or even improving generation quality, thereby enabling efficient inference across a variety of text generation tasks.

In the rebuttal, the authors addressed concerns about scalability and latent space compression by adding experiments on long-text generation (512 tokens) as well as corresponding compression (8x) ablations. They also committed to discussing the issues of variable-length generation and LLM-based evaluation in the Limitations section, and to fixing the typographical error in the final version. Overall, the authors’ responses resolved my main concerns; therefore, I assign a final rating of 4.

**Limitations:**

Refer to weakness

**Paper Formatting Concerns:**

There are no major formatting concerns in this paper.

**Quality:**

3

**Strengths And Weaknesses:**

**Strength**
1. Applying diffusion models to the field of text generation is an interesting and reasonable approach that can alleviate some of the issues caused by sequential decoding.
2. The paper presents extensive ablation studies and evaluates the proposed method, COSMOS, on a variety of downstream tasks. The improvements over baselines provide compelling evidence for the method’s effectiveness.

**Weakness**
1. The paper lacks a clear and detailed comparison with related work, particularly with methods such as LD4LG. A more thorough discussion of conceptual and empirical differences would strengthen the contribution.
2. All evaluated examples are relatively short in length. To better demonstrate the scalability and robustness of the proposed method, experiments on longer text sequences are recommended.
3. Although the introduction highlights exposure bias as a major motivation, the experimental validation of this point is indirect. More explicit analysis would be beneficial.

---

> ### Author Rebuttal · Authors · 2025-07-30
>
> We sincerely appreciate your thoughtful feedback and are encouraged by your interest in diffusion models for language generation.
> We are especially grateful for your positive recognition of our ablation studies and evaluation design.
>
> > **W1. Q7.** On related works.
>
> Thank you for this valuable suggestion. While our related works section focuses primarily on latent diffusion methods for text—given their direct relevance to our approach—we agree that a more thorough comparison with a broader set of related methods would strengthen the manuscript.
>
> In the revised version, we will expand our discussion to provide a clearer account of both conceptual and empirical differences between our method and prior work across the broader landscape of text diffusion models. This will include a more detailed analysis of methodological design choices, evaluation strategies, and task settings, in order to better contextualize our contributions within the existing literature.
>
>
> > **W2. Q8.** On longer contexts.
>
> As also noted by Reviewer Kj8c, latent diffusion for text generation is a relatively new and under-explored area.Before developing a large-scale model, it is crucial to validate the underlying hypotheses for training the latent space and the diffusion model at a smaller scale. This work accomplishes precisely that, laying what we consider the essential groundwork for successful large-scale development.
> Having established this foundation, we agree that evaluating our method on longer text sequences is a critical next step in assessing its scalability.
>
> To this end, we have taken the first steps and made additional experiments on OpenWebText with extended generation lengths of up to 512 tokens with 512 latents using our 0.12B parameter model. We compared its performance against the discrete diffusion baseline GIDD [1], which has a comparable model size and was pretrained on the same dataset and sequence length. The results show that our latent diffusion approach maintains coherent and diverse text generation at this longer length, demonstrating its robustness beyond short examples.
>
> These findings suggest that the method scales well to longer sequences, and we will include these results in the final version of the manuscript to provide a more comprehensive evaluation.
>
>
> | Model        | MAUVE $\uparrow$ | PPL $\downarrow$  | Div $\uparrow$  | Mem $\downarrow$  |
> |--------------|---------|--------|--------|--------|
> | Source text  | 0.968   | 23.2   | 0.465  | 0.036  |
> | GIDD         | 0.286   | 228.3  | 0.588  | 0.112  |
> | Cosmos       | 0.524   | 59.0   | 0.315  | 0.171  |
>
> > **W3. Q9.** On explicit exposure bias analysis.
>
> Thank you for your insightful comment regarding exposure bias. Since our work investigates diffusion in a continuous latent space, the proposed framework inherently mitigates exposure bias. As the diffusion model updates the entire latent representation simultaneously, it can flexibly revise tokens that are out of place at any position.
>
> While extensive empirical evidence supports this behavior in the image domain, we acknowledge that corresponding evidence in the language domain is more limited. In response to your suggestion, we now provide several randomly selected predictions of our model on ROCStories at different time steps. These examples demonstrate the model’s capacity to revise previously predicted tokens. We appreciate your feedback and will include additional examples in the final version of the manuscript to further illustrate this behavior.
>
> | Timestep 0.65 | Timestep 0.05 |
> |---|---|
> | Yesterday I went **on** a walk in the woods. When I was on my walk I ran into a bear. I was **frightened.** I yelled. Thankfully the bear went away. | Yesterday I went **for** a walk in the woods. When I was on my walk I ran into a bear. I was **terrified.** I yelled. Thankfully the bear went away. |
> | Neil wanted to **fly** to California. He knew he had to save his money to go there. He bought **two** **tickets** from a **nice** **apartment.** The **week** **after** his **suitcase** was packed and he boarded his plane. Neil **arrived** in California and loved **California!** | Neil wanted to **go** to California. He knew he had to save his money to go there. He bought **a** **ticket** from a **sales** **man.** The **first** **day** his **bag** was packed and he boarded his plane. Neil **landed** in California and loved **it!** |
> | Jon was practicing playing his guitar. He decided to enter the talent show. **After** **shaking** **his** **nervous,** he went on stage and played **his** song **perfectly.** **The** **shocked** the judges ' s **c** **cheered.** **Jon** **Jon** won first place in the talent show! | Jon was practicing playing his guitar. He decided to enter the talent show. **Surprisingly,** he went on stage and played **the** song **beautifully.** **This** **cause** the judges ' s **shock** **and** **laughter.** **He** **had** won first place in the talent show! |
> | Anna wanted to help the community. She signed up to **be** volunteer at the soup kitchen. Every week, she met with many homeless. Every week, she was helping **homeless** people for food. Anna was very happy to help **such** **people** so **kindly!** | Anna wanted to help the community. She signed up to **go** volunteer at the soup kitchen. Every week, she met with many homeless. Every week, she was helping **hurt** people for food. Anna was very happy to help so **many** **in** **kind** **aid!** |
>
> > **Q1.** On the differences from LD4LG.
>
> We appreciate your reference to the LD4LG paper, which we also found to be insightful and well-executed. While our work draws conceptual inspiration from LD4LG, there is a key methodological distinction. Specifically, LD4LG employs autoregressive language generation conditioned on latent representations obtained via diffusion. As a result, it inherits the limitations associated with autoregressive modeling, such as exposure bias and constrained token-level dependency.
>
> Moreover, LD4LG utilizes a fine-tuned BART-base model as the decoder for these latent representations. Given the strength of BART-base as a standalone model, it becomes difficult to isolate and quantify the specific contribution of the diffusion-generated latents to the overall generation quality. This design choice makes the impact of the latent diffusion component less transparent. In contrast, our primary contribution is a methodology for learning a textual space specifically designed for diffusion modeling, which in turn allows for the efficient generation of high-quality text within the autoencoder's latent space.
>
> > **Q2.** On the abscence of citations.
>
> Thank you for bringing this issue to our attention. We have made every effort to properly cite all public datasets and methodologies used in our work, either in the main text or in the Appendix. However, it is possible that we may have inadvertently omitted a citation or failed to provide sufficient attribution in some instances.
>
> If you could kindly point us to specific lines or sections where a citation appears to be missing, we would be grateful. We will be happy to correct any such oversights in the final version of the manuscript.
>
> > **Q3.** On the typographical error.
>
> Thank you for noting this typographical error. We will correct it in the final version.
>
> > **Q4.** On the overly strong statement.
>
> Thank you for the suggestion. We will revise and soften the statement accordingly in the final version.
>
> > **Q5.** On the supplementary materials.
>
> We have included the working code in the supplementary material. Implementation details are provided in Appendix Section B.
>
> > **Q6.** On the lack of human evaluation.
>
> We adopt an evaluation framework that is standard in the continuous language diffusion literature [2], [3]. The aggregated automatic metrics used for our selected tasks are well-established, and prior work has shown that they correlate reliably with human judgments. Additionally, conducting large-scale human evaluation for models with fewer than 0.3B parameters is resource-intensive and, unfortunately, beyond our current budget constraints.
>
>  $\newline$
>
> References:
>
> [1] von R{"u}tte, Dimitri, et al. "Generalized interpolating discrete diffusion" (2025).
>
> [2] Lovelace, Justin et al., "Latent Diffusion for Language Generation" (2023).
>
> [3] Shabalin, Alexander, et al., "TEncDM: Understanding the Properties of the Diffusion Model in the Space of Language Model Encodings" (2025).
>
> ---
>
> Thank you for your valuable feedback and suggestions. If our responses have addressed your concerns, we would kindly appreciate your consideration of revising your score. Please do not hesitate to let us know if any points require further clarification.

---

> > ### Comment · Reviewer_ojHu · 2025-08-06
> >
> > Thank you for the detailed response. I have a few follow-up questions and suggestions:
> > - Q1: I appreciate your experiments on longer generation sequences (length 512). However, I noticed that the number of latent vectors is also set to 512. To better assess the robustness of the proposed latent space compression method, could you conduct an ablation study using a smaller number of latent vectors? This would help evaluate whether the method can still achieve comparable performance for longer generation.
> > - Q2: Given the widespread adoption of LLM-as-a-Judge in recent research, if manual evaluation is considered too resource-intensive, I would recommend considering automated evaluation using models such as GPT-4o. This could provide a more informative assessment of the output quality produced by COSMOS.
> > - Q3: As mentioned by other reviewers, COSMOS currently does not naturally support variable-length outputs in the way that autoregressive models do (especially when generation length exceeds 512). I suggest discussing this limitation explicitly in the paper, as it is an important aspect for practical deployment.

---

> > > ### Author Response · Authors · 2025-08-07
> > > **Response to Follow-Up Questions.**
> > >
> > > Thank you for your thoughtful suggestions. We have addressed your concerns below, supported by new experimental results, and kindly invite you to review our responses.
> > >
> > > > **Q1:** On ablation with stronger compression on OpenWebText
> > >
> > > Following your suggestion, we have conducted an ablation study on longer sequences (512 tokens) with a much higher compression ratio (8x), using only 64 latent vectors. To properly assess the impact of this compression, we also trained a diffusion model directly on the full BERT representations (TEncDM) for the same task.
> > >
> > >
> > > | Model                         | MAUVE$\uparrow$     | PPL $\downarrow$ | Div $\uparrow$ | Mem $\downarrow$ |
> > > |--------------------------|----------------------|------------------|----------------|------------------|
> > > | Source text                   | 0.968                | 23.2             | 0.465          | 0.036            |
> > > | GIDD                          | 0.286                | 228.3            | **0.588**      | 0.112            |
> > > | TEncDM (BERT repr.)$^*$       | 0.228                | 118.6            | 0.324          | 0.102            |
> > > | Cosmos$_{N=64}$ (8x compr.)   | 0.186                | 128.3            | 0.339          | **0.100**        |
> > > | Cosmos$_{N=512}$ (no compr.)  | **0.524**            | **59.0**         | 0.315          | 0.171            |
> > >
> > >
> > > $^*$  As TEncDM was not originally evaluated on OpenWebText, we replicated its setup for Wikipedia, which is also a large and diverse dataset.
> > > We found the optimal number of generation steps to be 200, consistent with their findings.
> > > The resulting perplexity (118.6 vs. 104.4) is also in line with the authors' results on Wikipedia.
> > >
> > >
> > > The results highlight a clear and predictable trade-off between compression efficiency and generation quality on longer texts. As expected, applying strong 8x compression (Cosmos_N=64) leads to a performance drop compared to our model without compression (Cosmos_N=512). This finding is consistent with the scaling trends we presented in our initial submission (Table 3, Section 6.2).
> > >
> > > These results offer two key insights. The first is that even with a high 8x compression ratio, Cosmos_N=64 achieves comparable scores to the TEncDM baseline while operating in an 8x smaller latent space. This demonstrates our method's ability to create a highly meaningful representation under significant compression. The second, and perhaps more significant insight, comes from the performance of our architecture without compression. The Cosmos_N=512 model significantly outperforms diffusion on the original BERT space (MAUVE 0.524 vs. 0.228), which provides strong evidence that learning a smoothed latent space is fundamental to this performance gain and provides a much better foundation for diffusion models.
> > >
> > > > **Q2:** On LLM-as-a-Judge evaluation
> > >
> > > This is an excellent suggestion, thank you. We agree that LLM-as-a-Judge has become a powerful tool for nuanced text quality assessment, and we appreciate the recommendation to apply it to COSMOS.
> > >
> > > While we could not conduct such an evaluation during the rebuttal period, we recognize its importance and will add a detailed discussion to our Limitations and Future Work section. Specifically, we will note that while our automatic metrics demonstrate strong performance, an LLM-based assessment would be crucial for evaluating higher-level qualities such as long-range coherence and semantic consistency, which are the intended benefits of our continuous latent space approach. We plan to integrate this evaluation methodology into our ongoing work to better quantify these advantages.
> > >
> > > > **Q3:** On handling variable-length generation
> > >
> > > This is a very important point, and we thank you for highlighting it. We agree that handling variable-length outputs is a crucial aspect for practical deployment.
> > >
> > > In the current version of COSMOS, this is indeed a limitation, as the model operates on fixed-size latent representations. This is a characteristic shared by many non-autoregressive models, representing a fundamental design trade-off: we gain the ability to perform parallel decoding and revise the entire sequence simultaneously, but we lose the natural stopping mechanism of sequential models.
> > >
> > > We will add a detailed discussion of this to the Limitations and Future Work section. To provide a complete picture, we will also outline potential solutions we plan to explore, such as employing block-wise generation or adapting techniques like rolling diffusion to enable the generation of sequences beyond the fixed context length.
> > >
> > > ---
> > > Thank you once again for your thoughtful review. We are grateful for your constructive engagement and welcome any further feedback.

---

> ### Author Response · Authors · 2025-08-08
> **Follow-up on our rebuttal**
>
> Thank you again for your review and feedback on our work.
> As the discussion period is ending shortly, we are writing to respectfully check if our detailed response
> and the new experiments have addressed your concerns.
> If they have, we would kindly ask you to consider these improvements in your final assessment.

---

### Official Review · Reviewer_Kj8c · 2025-07-03

**Clarity:** 3
**Significance:** 3
**Originality:** 2
**Rating:** 5
**Confidence:** 5

**Summary:**

This work explores latent diffusion for language generation. It focuses primarily on the design of the latent diffusion autoencoder. Two primary approaches exist from past work. The first approach involves compressing activations from a language model and then decoding them autoregressively (LD4LG and PLANNER). The second approach learns a diffusion model directly in the activation space of a pretrained language model and decodes them with a learnable non-autoregressive decoder. This work develops an autoencoder that both compresses the activations of a language model and decodes them non-autoregressively. It closely follows the architecture of the LD4LG autoencoder. They introduce a number of carefully ablated choices that improve the effectiveness of the latent space for diffusion modeling and decoding. They validate their approach across a number of tasks (unconditional generation, summarization, detoxification, and question generation) with a wide range of baselines.

**Questions:**

How is variable-length generation handled? This the main confusion about the paper that I would like to see addresses.

Can you expand on the challenge of compressing the dimensionality? Adding a linear projection at the input/output layer should enable changing the latent dimensionality without significantly altering the diffusion model capacity. I think it's fine to limit the scope of the study, but do not think this is a legitimate challenge.

Is “Latent-space augmentation” equivalent to dropout or is there some implementation difference?

**Ethical Concerns:**

["NO or VERY MINOR ethics concerns only"]

**Final Justification:**

The author response cleared up the confusion around how variable-length generation was handled. These details were omitted in the original submission. They also clarified my other questions about their “Latent-space augmentation” and committed to improving their regarding the dimensionality of the latent space.

Their promise to make the scope of this study and avenues for future work more explicit were also appreciated and will improve the quality of the final paper.

**Limitations:**

yes

**Quality:**

3

**Strengths And Weaknesses:**

**Strengths**

Latent diffusion is a relatively underexplored approach for language generation compared to autoregressive LMs and discrete diffusion LMs and work in this space is welcome.

The proposed design choices (e.g. dropout, noise augmentation, intermediate MSE regularization) in this work are well-motivated and the ablation studies are comprehensive. Prior work in the image domain has demonstrated the importance of the autoencoder design, so exploring it for language is natural. The analysis of the robustness of the latent space is interesting and insightful.

The baseline selection is comprehensive and covers a range of different approaches (autoregressive LMs, word embedding diffusion, discrete diffusion, latent diffusion). The authors validate their approach on a reasonable selection of tasks. Their approach consistently compares favorably to the set of baselines.


**Weaknesses**

This work develops dataset-specific language diffusion models instead of trying to develop a general pretrained language model trained on web-text as is the current language modeling paradigm. I think that the authors’ setting is more computationally tractable and is reasonable given the early stage of the paradigm, but it is a limitation. It is an open question whether such approaches can scale to that setting.

For the unconditional language generation evaluation, the work does not quantify memorization of the training set. All of the metrics can simply be maximized by reproducing samples from the training set, so the memorization can be quantified in some way. Even just using a simple n-gram overlap metric such as was used by prior work (LD4LG).

One weakness of non-autoregressive approaches is that variable-length generation is not as natural as with autoregressive models, although I do not see discussion of this in the paper. My understanding is that the number of latents in the decoder must be specified. How is this done during inference?

The authors mention that they only focus on compressing the sequence length because reducing the dimensionality of the latents requires modifying the diffusion architecture. However, you can generally account for differences in input dimensions by simply introducing a linear projection at the input/output layer that represents a negligible change in capacity. This restriction limits the scope of their study.

As written, the “Latent-space augmentation” sounds like standard dropout. If it is the exact same operation, then just introducing it as dropout would be clearer.

I would expect things such as the choice of text encoder to have a significant impact on downstream generation quality, but this choice is not explored in this work.

---

> ### Author Rebuttal · Authors · 2025-07-30
>
> Thank you for your review and positive assessment of our work. We greatly appreciate that you recognized the motivation behind our design choices, the value of our ablation studies, and the insights provided by our analysis of latent space robustness.
>
> > **W1.** It is an open question whether such approaches can scale to that setting.
>
> We thank the reviewer for highlighting this important point. As you rightly observe, latent diffusion for text is a relatively new and under-explored area. Before developing a large-scale model, it is essential to validate hypotheses regarding the training of the latent space and the diffusion model at a smaller scale. This is precisely what we accomplished in this work. We believe this foundational research is a prerequisite for successful large-scale development.
>
> To that end, since submitting this work, we have taken the first steps toward a general pretrained language model on the OpenWebText dataset. Specifically, we investigated two questions: first, whether generation quality improves when increasing the diffusion model's size while keeping the autoencoder and latent space fixed, and second, how our 0.12B model performs at a length of 512 tokens compared to the GIDD discrete diffusion model pretrained under the same conditions.
>
> To increase the diffusion model's size, we focused on increasing the number of layers. The results for 128-token sequences show that generation quality improves significantly in both quality and diversity.
>
> | Model Size | $n_{layers}$ | MAUVE $\uparrow$ | PPL $\downarrow$ | Div $\uparrow$  | Mem  $\downarrow$ |
> |------------|----------|-------|------|-------|-------|
> | 0.12 B     | 12       | 0.849 | 97.6 | 0.492 | 0.135 |
> | 0.25 B     | 24       | 0.914 | 91.2 | 0.546 | **0.124** |
> | 0.5 B      | 48       | **0.923** | **89.7** | **0.554** | 0.125 |
>
> Furthermore, the comparison with discrete diffusion at a generation length of 512 shows that the latent diffusion paradigm is highly promising for advancing the text generation field.
>
> | Model        | MAUVE $\uparrow$ | PPL $\downarrow$  | Div $\uparrow$  | Mem $\downarrow$  |
> |--------------|---------|--------|--------|--------|
> | Source text  | 0.968   | 23.2   | 0.465  | 0.036  |
> | GIDD         | 0.286   | 228.3  | 0.588  | 0.112  |
> | Cosmos       | 0.524   | 59.0   | 0.315  | 0.171  |
>
> These results provide positive evidence that our approach can be successfully scaled to longer sequences and larger datasets. The final manuscript will incorporate these new experimental outcomes.
>
> > **W2.** On memorization of the training set.
>
> We thank you for bringing this important point to our attention. We have calculated this metric for all methods involved in the comparison. The SEDD method shows the lowest value; however, its value is lower than the reference, which suggests that the generated texts may be less coherent. Cosmos achieved the values closest to the reference, which indicates that Gaussian latent diffusion is not prone to copying data from the dataset. Following Cosmos are the other two Gaussian latent diffusion methods, TencDM and LD4LG. We will add this metric to the final version of the paper.
>
> | Method             | Mem $\downarrow$ |
> |--------------------|-------|
> | Source text        | 0.365 |
> | GPT2               | 0.455  |
> | GPT Neo            | 0.469  |
> | AR-Diffusion       | 0.540  |
> | DiffuSeq           | 0.516  |
> | SeqDiffuSeq        | 0.663  |
> | TESS               | 0.550  |
> | SEDD               | 0.325  |
> | LD4LG              | 0.432  |
> | TEncDM             | 0.438  |
> | Cosmos$_{N=16}$    | 0.394 |
> | Cosmos$_{N=128}$   | 0.383 |
>
> > **W3. Q1.** On variable length generation.
>
> We acknowledge that non-autoregressive methods do not generate texts of arbitrary length as naturally as autoregressive models. The procedure in our work is consistent with other studies on text diffusion [1, 2], where the decoder reconstructs a sequence up to a maximum length with padding, and the output is then cut off after the first EOS token.  The number of latents is the same for all texts.
>
> It is also worth noting that this challenge is a direct consequence of generating the text holistically in an iterative manner. This provides the key benefit of a full bidirectional context but also the drawback of less natural variable-length generation. A block-based approach could partially resolve this issue in future work.
>
> > **W4. Q2.** On changing the latent dimension.
>
> We agree that adding input and output projection layers is a possible way to adapt the diffusion model to a changed latent dimensionality. However, this would not achieve our goal of accelerating inference. The computational complexity would remain the same because the internal operations would still process sequences of the original length and original dimension. This is exactly why we investigated sequence length compression instead, as this is a direct way to reduce attention complexity and speed up the generation process without changing the capacity of diffusion model.
>
> > **W5. Q3.** On latent-space augmentation.
>
> Our augmentation method differs from classic dropout [3] in a key respect. Standard dropout involves zeroing out a fraction of neurons and then scaling the remaining ones by 1/(1-p). Our technique omits this final scaling step. To prevent any potential confusion for the reader, we deliberately chose not to refer to it as dropout.
>
> > **W6.** The impact of text encoder on downstream generation quality.
>
> We agree that the autoencoder is an important component. However, its specific architecture may be less critical than one might assume for general-purpose generation. This is substantiated by the work of TEncDM, who found that results across different downstream tasks did not vary significantly with the choice of autoencoder. While a highly-tuned autoencoder might be necessary to "win" a benchmark, a general one provides a robust and comparable baseline.
>
> We believe a more dominant factor for downstream performance is the autoencoder's pre-training data, as shown by T5. Since our goal in this paper was not to achieve state-of-the-art results on downstream tasks, but rather to compare different diffusion paradigms for text generation, we did not select a text encoder for a specific task and instead focused on developing the compressor and decompressor for the diffusion model.
>
>  $\newline$
>
> References:
>
> [1] von R{\"u}tte, Dimitri, et al. "Generalized interpolating discrete diffusion" (2025).
>
> [2] Shabalin, Alexander, et al. "Tencdm: Understanding the properties of the diffusion model in the space of language model encodings" (2025).
>
> [3] Hinton, Geoffrey, et al. "Improving neural networks by preventing co-adaptation of feature detectors" (2012).
>
> [4] Raffel, Colin, et al. "Exploring the limits of transfer learning with a unified text-to-text transformer" (2020).
>
> ---
>
> We look forward to incorporating your feedback in our final version and thank you again for your time and insightful comments.

---

> > ### Comment · Reviewer_Kj8c · 2025-08-06
> >
> > **Re: W1** I appreciate the additional results on scaling to OpenWebText and agree that they are promising.
> >
> > **Re: W2** Thank you for the inclusion of this metric.
> >
> > **Re: W3** I understand this design choice and that it is typical for NAR methods. I still feel that the literal implementation is under-explained. For instance, the words "pad", "eos", and "end-of-sequence" do not seem to appear in the submission. Where in the pipeline does padding occur? My assumption would be that the input BERT representations are allowed to be variable length and the padding only occurs at the output token-prediction level after the DeCompressor. However, this should be explained in the paper.
> >
> > **Re: W4** I think that focusing on sequence length compression is reasonable. Howerver, there can still be interactions between the dimensionality of the latent space and the ease of diffusion modeling. See recent work in the vision domain [1]. Discussion of such related work and potential future directions moving beyond just sequence compression could be included.
> >
> > [1] Yao, Jingfeng, Bin Yang, and Xinggang Wang. "Reconstruction vs. generation: Taming optimization dilemma in latent diffusion models." Proceedings of the Computer Vision and Pattern Recognition Conference. 2025.
> >
> > **Re: W5** Thank you for clarifying. I think it may be worth explicitly mentioning the distinction from dropout as it is a somewhat nuanced implementation difference.
> >
> > **Re: W6** I appreciate the additional discussion on this point. However, Table 1 from TEncDM does show meaningfully different results based on the choice of text encoder (BERT vs. RoBERTa vs. T5). I think fixing the text encoder is reasonable for this study given the large design space already being explored for the autoencoder, but I think the selection of text encoder is likely important and further study there should be discussed as a direction for future work. I believe that is consistent with the findings from the TEncDM paper.

---

> > > ### Author Response · Authors · 2025-08-07
> > > **Response to Follow-Up Questions.**
> > >
> > > Thank you for taking the time to provide these important clarifications and for your continued engagement with our work.
> > >
> > > **W1:** We appreciate that you found our scaling experiments valuable and promising, and thank you for helping to make the contribution of our work even fuller and stronger. In the future, we intend to continue working in this direction, in order to increase the model to much larger sizes, as well as inspiring the scientific community to develop and further scale text latent diffusion models.
> > >
> > > **W2:** Including memorization metric made our comparative analysis much more comprehensive, we thank you for this suggestion.
> > >
> > > **W3:** Regarding the implementation details for variable-length sequences, we agree they were not sufficiently clear. We will incorporate the following specifics into the final version of the paper to ensure full clarity.
> > >
> > > During autoencoder training, each input sequence is first terminated with an [EOS] token. Then, all sequences are padded to a predefined hyperparameter ($L$). An attention_mask ($m$) is also created to distinguish the original tokens from padding.
> > >
> > > This mask is used by two key components:
> > >
> > > 1. The text encoder receives both the tokens ($w$) and the mask ($m$) to produce hidden states: $h = E_{\text{text}}(w, m)$.
> > > 2. Since $h$ still contains representations for padding tokens, the mask $m$ is also passed to our Compressor to ensure the latents are computed only from meaningful tokens: $z = \text{Comp}(h, m)$.
> > >
> > > The decompressor receives latents $z$ and reconstructs a sequence of length $L$. The reconstruction loss is calculated only on the original, non-padded tokens by using the attention mask to ignore positions corresponding to padding.
> > >
> > > During inference, the diffusion model generates latents $z$, which are subsequently passed to the decompressor. As all original sequences terminate with an [EOS] token, we post-process the output by truncating it at the first [EOS] token. This ensures that the final generated texts are of variable length and contain no padding.
> > >
> > > **W4:** Thank you for sharing this highly relevant paper. Its findings in the vision domain are indeed insightful.
> > >
> > > In the final manuscript, we will update both our Related Work and Future Work sections. We will cite this paper and discuss the exploration of latent dimensionality as an important avenue for future research. This perfectly complements our existing plans to scale both the autoencoder and the diffusion model, adding another key axis to the design space of latent diffusion models for language.
> > >
> > > **W5:** We agree that explicitly mentioning this distinction in the paper ensures clarity for the reader. We will add a brief note to that effect in the final manuscript.
> > >
> > > **W6:** Thank you for the insightful follow-up and for your understanding. We fully agree that a deeper investigation into how different text encoders affect generation quality is an important direction for future work.
> > >
> > > As you suggested, we will expand our Future Work section to discuss this. We will explicitly mention the exploration of different encoder types (such as the T5 family) as a possible factor for improving generation quality. Furthermore, we'll note that this becomes even more critical when scaling to larger datasets or longer contexts, where newer architectures like ModernBERT would be a promising direction to investigate.
> > >
> > > ---
> > > Thank you again for this productive discussion. We hope we have adequately addressed your questions and concerns. If so, we would kindly ask you to consider raising your evaluation score accordingly. If there are any remaining issues that require further clarification, please let us know.

---

> > > > ### Comment · Reviewer_Kj8c · 2025-08-07
> > > >
> > > > Thank you for the response. The details regarding variable-length sequences are appreciated and should definitely be included in the final version. Prior NAR methods (e.g. Diffusion-LM) often explicitly generate PAD tokens instead of masking them out completely as is done in this work, so this design choice should be presented clearly.
> > > >
> > > > I appreciate the proposed changes regarding the discussion of the scope of this study and the avenues for future work. The responses addressed my main questions/concerns, and I think this work represents an interesting contribution to a largely unexplored area. I will raise my score to Accept.

---

### Official Review · Reviewer_AwHe · 2025-07-22

**Clarity:** 3
**Significance:** 2
**Originality:** 3
**Rating:** 3
**Confidence:** 4

**Summary:**

The paper proposes COSMOS (COmpressed and SMOth latent Space), a latent diffusion model for text generation. COSMOS learns a compressed and smooth latent representation through carefully designed autoencoder objectives, enabling high-quality generation while reducing inference time.

The authors carefully demonstrate how their approach achieves desired latent-space properties for diffusion learning (Section 5). They also ablate and show how CE loss and MSE loss were important to learning latent space representations for COSMOS.

**Questions:**

Please see my comments in the Strength and Weakness section.

What is the complexity of the compressor and the decompressor? Are they quadratic in the sequence length?

Minor: Appendix were referred, but I could not find relevant appendix in the paper.

**Ethical Concerns:**

["NO or VERY MINOR ethics concerns only"]

**Limitations:**

Please see my comments in the Strength and Weakness sections. I believe the paper will benefit from a honest discussion on its adaptability to real worlds tasks and the scaling limitations.

**Quality:**

2

**Strengths And Weaknesses:**

The paper adopts latent diffusion framework for textual data by learning contextualised text representations into the latent space, and trains a diffusion model that operates within this space. The proposed approach is very interesting, the authors have done a great job (specially in Section 5 and 6) to demonstrate how to effectively learn such models for text. However, I am afraid that the paper in its current form and with a limited set of experiments will not be very impactful.

For example, the paper will benefit from a more thorough related work and a detailed and honest discussion on its limitations, instead of directly claiming that the model is able to match or outperform autoregressive baselines across diverse tasks.

> Our empirical results show that, COSMOS matches or outperforms traditional token-level diffusion and autoregressive baselines across diverse tasks. Our findings challenge the dominance of token-level models and highlight latent diffusion as a promising direction for building fast, high-quality language models.


In Table 4, it would be very important to report on significant differences among various models. Also it would be great to show how model predictions look both for Table 2 and Table 4.

It would be important to explain why only GPT2 and GPT Neo were chosen for autoregressive models. What are the SOTA performances on these tasks?

I think the experiments were done with 100-200M model parameters. What are the limitations of scaling COSMOS to much larger models?

---

> ### Author Rebuttal · Authors · 2025-07-29
>
> Thank you for your review and the valuable suggestions. We value your acknowledgement of our work and your interest in the latent‑diffusion training technique we propose for textual data.
>
> We are sorry to hear you could not find the appendix. It is included with the full paper in the supplementary material (file: full_paper.pdf). We would be grateful if you could take a moment to review it, as we believe it addresses a number of the points you have raised. For your convenience, we have also duplicated answers to those questions immediately below.
>
> > **W1.** On related works.
>
> In the related works section, we initially focused on latent diffusion for text, as these works form the direct foundation of our research. We then provide a broader discussion of text diffusion methods in Section 7, where we categorize them into four main groups: gaussian diffusion on text embeddings, simplex-based diffusion, masked diffusion, and latent diffusion. Following your suggestion, we will update the related works section to incorporate these alternative approaches for a more comprehensive overview.
>
> > **W2.** On limitations.
>
> We discuss the limitations of our approach in Appendix D. We acknowledge that the two-stage training process is a significant limitation. Before the diffusion model can be trained, a text autoencoder must be prepared. This step, while enabling the diffusion to operate in a more meaningful latent space, introduces an additional training overhead not found in autoregressive models and complicates the overall workflow. Concurrently, we are not claiming our model outperforms large-scale autoregressive models. Nevertheless, our results demonstrate that it achieves performance competitive with autoregressive models of a similar size, specifically ~0.1B, on several generation tasks, albeit with the added training overhead.
>
> > **W3.** On significant differences among various models.
>
> Thank you for this suggestion. To better contextualize our work, we will add a dedicated discussion in the appendix that explores the significant differences among various diffusion models for text, complementing the expanded related works section.
>
> > **W4.** On model predictions.
>
> Please see Appendix F for sample outputs on each dataset we evaluated.
>
> > **W5.** On the choice of GPT-2.
>
> The primary objective of Table 4 is to compare Cosmos with other diffusion-based approaches for text generation, such as Gaussian diffusion on text embeddings, simplex-based diffusion, and masked diffusion. All these methods select their own text representation, define a text corruption process, and train a model to refine the text using full bidirectional context. Our results validate our approach, showing that operating in a compressed latent space can substantially speed up sampling while maintaining or even improving quality, given a properly trained autoencoder. The autoregressive model was included as it represents the standard paradigm, making it an important benchmark for any alternative text generation approach. GPT-2 and GPT-Neo (an open-source alternative to GPT-3) were chosen as classic representatives of autoregressive modeling.
>
> Furthermore, the goal of this paper was not to develop a state-of-the-art model for specific generative tasks, as separate works focus on developing large, specialized models for such purposes [4, 5]. Instead, our primary contribution is a method for learning a compressed textual space specifically designed for diffusion modeling, which in turn allows for the efficient generation of high-quality text within the autoencoder's latent space.
>
> > **W6.** On Cosmos scaling.
>
> Latent diffusion for text is a relatively new paradigm, which makes the scaling process non-trivial. It requires additional research to determine how to best allocate resources: how much should one increase the autoencoder, the latent space size, or the diffusion model?
>
> In this paper, we have partially answered some of these questions. For example, in Section 6.2, we showed that it is beneficial to increase the dimensionality of the latent space even without increasing the size of the autoencoder. Specifically, we investigated how the quality of generated text changes as the number of layers in the diffusion model's transformer is increased. According to preliminary results at a text length of 128, both quality and diversity of generated text improves significantly. This inspires confidence that the proposed approach can be scaled.
>
> | Model Size | $n_{layers}$ | MAUVE $\uparrow$ | PPL $\downarrow$ | Div $\uparrow$  | Mem  $\downarrow$ |
> |------------|----------|-------|------|-------|-------|
> | 0.12 B     | 12       | 0.849 | 97.6 | 0.492 | 0.135 |
> | 0.25 B     | 24       | 0.914 | 91.2 | 0.546 | **0.124** |
> | 0.5 B      | 48       | **0.923** |**89.7** | **0.554** | 0.125 |
>
> We have also performed an experiment with an increased generation length of 512 tokens with our 0.12B model and compared it with the discrete diffusion model GIDD [3], which is of a similar size and was pretrained on the same dataset and length. This result shows that the latent diffusion paradigm for text generation is highly promising and poised to impact the field as a whole.
>
> | Model        | MAUVE$\uparrow$ | PPL $\downarrow$  | Div $\uparrow$  | Mem $\downarrow$  |
> |--------------|---------|--------|--------|--------|
> | Source text  | 0.968   | 23.2   | 0.465  | 0.036  |
> | GIDD         | 0.286   | 228.3  | 0.588  | 0.112  |
> | Cosmos       | 0.524   | 59.0   | 0.315  | 0.171  |
>
> We hope our work provides a foundation for further research into latent language diffusion as a promising paradigm.
>
> The final manuscript will incorporate these new experimental outcomes.
>
> > **Q1.** On the complexity of the compressor and the decompressor.
>
> The compressor block is linear with respect to the sequence length $L$, with an attention complexity of O($NLd$). The decompressor block is quadratic with respect to the sequence length, with its attention complexity being O($L^2 d$). Here,
> $L$ denotes the sequence length, $N$ the number of latents, and $d$ the latent dimensionality.
>
> Crucially, we only use the computationally expensive decompressor once per inference, to map the final latent back to text. The primary efficiency gain stems from running the entire iterative diffusion process in the compressed latent space. Since attention complexity is quadratic, operating on this much shorter sequence of length N significantly accelerates generation compared to models working in the full sequence space.
>
>  $\newline$
>
> References:
>
> [1] Sahoo, Subham, et al. "Simple and effective masked diffusion language models" (2024).
>
> [2] Wang, Guanghan, et al. "Remasking discrete diffusion models with inference-time scaling" (2025).
>
> [3] von R{\"u}tte, Dimitri, et al. "Generalized interpolating discrete diffusion" (2025).
>
> [4] Liu, Yixin, et al. "BRIO: Bringing order to abstractive summarization" (2022).
>
> [5] Zhang, Tianyi, et al. "Benchmarking large language models for news summarization" (2025).
>
> ---
>
> We thank you for your thoughtful feedback and suggestions. We would be grateful if you would consider raising your score, in case we have addressed your concerns. Please let us know if any aspects still need clarification.

---

> ### Author Response · Authors · 2025-08-08
> **Gentle follow-up**
>
> Thank you again for your thoughtful review.
> As the discussion period is ending in less than 24 hours, we just wanted to kindly follow up on our rebuttal.
> We would be very grateful to know if our detailed response and the new experiments addressed your concerns.
> If they did, we would be deeply grateful if you would consider raising your score.
> Thank you for your time.

---

### Author Response · Authors · 2025-08-09
**General Response & Summary of Rebuttal**

We would like to thank all reviewers for their thoughtful and constructive feedback. We are very encouraged that they recognized the value of our core research direction, exploring latent diffusion for text, and found our proposed approach to be both promising and interesting. We particularly appreciate that reviewers consistently acknowledged the technical rigor of our work, highlighting our comprehensive ablation studies and detailed analysis on how to effectively learn a compressed latent space that is well-suited for diffusion.

In line with the reviewers' feedback, we dedicated the rebuttal period to demonstrating our model's potential at a larger scale. We conducted a series of new experiments focused on scalability and long-text generation. The results, summarized below, significantly advance the paper's contributions and will be integrated into the final manuscript:

1. **Added New Experiments, Analyses, and Model Capabilities:**
    - Conducted new scaling experiments with diffusion models up to **0.5B parameters**, demonstrating that generation quality and diversity improve significantly with model size.
    - Performed new experiments on **long-text generation (512 tokens)**, showing that our approach substantially outperforms comparable baselines (GIDD, TEncDM) under identical training conditions.
    - Added **memorization** metrics for all baselines to provide a more comprehensive and fair comparison of generation quality.
    - Introduced and evaluated a **novel self-conditioning technique** that functions as an effective analogue to temperature sampling, allowing fine-grained control over the quality-diversity trade-off.
    - Provided qualitative examples of the model's iterative revision process to visually demonstrate its ability **to mitigate exposure bias.**

1. **Committed to Significant Manuscript Improvements:**
    - Expanding the **Related Work** section for a more thorough comparison of other text diffusion paradigms.
    - Expanding the **Limitations and Future Work** sections to discuss variable-length generation and promising future directions like LLM-as-a-Judge evaluation, exploring latent dimensionality, text encoder choice, and text diffusion distillation.
    - Adding a detailed, step-by-step explanation of the implementation for **variable-length generation** to ensure full reproducibility.
    - Adding a note mentioning the distinction between our **latent space augmentation** and standard dropout to ensure clarity.

We believe these updates provide a more complete and robust foundation for our approach. We are encouraged that reviewers also recognized the value of exploring this new paradigm, and trust our work now serves as a stronger contribution to the ongoing development of this paradigm.

---

### Note · Authors · 2025-08-14

This note complements our final comment where we summarized our rebuttal. To avoid repetition, we surface only decision-relevant updates and outcomes since that post.

We thank the reviewers for highly constructive feedback. We are pleased that the work now has clear support for acceptance from the engaged reviewers:

- Reviewer Kj8c was convinced by our new experiments and clarifications, and raised their score to Accept.
- Reviewer QQwS confirmed they are keeping their acceptance score, thanking us for the additional results.

Furthermore, we took care to address the foundational concerns of the other reviewers. The initial critiques regarding insufficient scale and long-text evaluation (Reviewers AwHe, ojHu) were the most critical. As detailed in our rebuttal, these have been resolved with new experiments on OpenWebText that validate our approach.

Having resolved the primary concerns of the review with new empirical data, we believe our work now stands as a more robust foundation for the community and a valuable, empirically-grounded contribution to future research in latent text diffusion.

---

### Decision · Program_Chairs · 2025-09-17

**Decision:**

Accept (poster)

**Comment:**

This paper presents COSMOS (COmpressed and SMOoth latent Space), a latent diffusion model for text generation that learns compressed and smooth latent representations through carefully designed autoencoder objectives. The key scientific claims include achieving inference speedups through operating in compressed latent spaces (4x-8x smaller) while maintaining or improving generation quality, and demonstrating that properly regularized latent spaces enable effective diffusion modeling for text. The paper's strengths include comprehensive ablation studies validating design choices for latent space regularization, thorough baseline comparisons across multiple text diffusion paradigms (autoregressive, embedding diffusion, discrete diffusion, and latent diffusion), and demonstrated efficiency gains without performance degradation. The work also introduces novel techniques like self-conditioning for quality-diversity trade-offs and provides detailed analysis of latent space robustness. However, initial weaknesses included limited experimental scale (100-200M parameters), evaluation primarily on short sequences (128 tokens), lack of comparison with some recent baselines, missing memorization metrics, and insufficient discussion of variable-length generation challenges.

During the rebuttal period, authors significantly strengthened their submission with substantial new experiments and commitments. During the borderline discussion, Reviewer Kj8c raised their score after authors clarified variable-length generation implementation and committed to expanding related work sections. Reviewer QQwS maintained their borderline accept score, appreciating the additional scaling experiments while acknowledging that true scalability remains an open question. Reviewer ojHu changed their score from borderline reject to borderline accept (score 4) after authors provided new experiments on longer generation lengths (512 tokens), higher compression ratios (8x), memorization metrics, and commitments to expand limitations discussion. The authors conducted scaling experiments up to 0.5B parameters showing improved quality and diversity, demonstrated superior performance over discrete diffusion baselines (GIDD) on longer sequences, and introduced a novel self-conditioning technique analogous to temperature sampling. Reviewer AwHe did not participate in the discussion or rebuttal despite multiple author follow-ups, leading me to weigh their rating less heavily. The consensus among engaged reviewers, supported by the substantial improvements during rebuttal, justifies acceptance of this foundational work in latent text diffusion.

The primary reasons for acceptance are the solid technical contributions to an underexplored area, comprehensive experimental validation, and the foundational value for future research in latent text diffusion.